# Chromatin mapping identifies BasR, a key regulator of bacteria-triggered production of fungal secondary metabolites

Juliane Fischer[1,2†], Sebastian Y Müller[3†‡], Tina Netzker[1†], Nils Jäger[4†], Agnieszka Gacek-Matthews[5,6], Kirstin Scherlach[7], Maria C Stroe[1,2], María García-Altares[7], Francesco Pezzini[3§], Hanno Schoeler[1,2], Michael Reichelt[8], Jonathan Gershenzon[8], Mario KC Krespach[1,2], Ekaterina Shelest[3§], Volker Schroeckh[1], Vito Valiante[9], Thorsten Heinzel[4], Christian Hertweck[7,10], Joseph Strauss[5*], Axel A Brakhage[1,2*]

[1]Department of Molecular and Applied Microbiology, Leibniz Institute for Natural Product Research and Infection Biology, Jena, Germany; [2]Institute of Microbiology, Friedrich Schiller University Jena, Jena, Germany; [3]Systems Biology and Bioinformatics, Leibniz Institute for Natural Product Research and Infection Biology, Jena, Germany; [4]Department of Biochemistry, Friedrich Schiller University, Jena, Germany; [5]Department for Applied Genetics and Cell Biology, BOKU University of Natural Resources and Life Sciences, Vienna, Austria; [6]Institute of Microbiology, University of Veterinary Medicine, Vienna, Austria; [7]Department of Biomolecular Chemistry, Leibniz Institute for Natural Product Research and Infection Biology, Jena, Germany; [8]Department of Biochemistry, Max Planck Institute for Chemical Ecology, Jena, Germany; [9]Leibniz Research Group – Biobricks of Microbial Natural Product Syntheses, Leibniz Institute for Natural Product Research and Infection Biology, Jena, Germany; [10]Chair for Natural Product Chemistry, Friedrich Schiller University, Jena, Germany

*For correspondence:
joseph.strauss@boku.ac.at (JS);
axel.brakhage@hki-jena.de (AAB)

[†]These authors contributed equally to this work

Present address: [‡]Department of Plant Sciences, University of Cambridge, Cambridge, United Kingdom; [§]Bioinformatics Unit, German Centre for Integrative Biodiversity Research, Leipzig, Germany

**Abstract** The eukaryotic epigenetic machinery can be modified by bacteria to reprogram the response of eukaryotes during their interaction with microorganisms. We discovered that the bacterium *Streptomyces rapamycinicus* triggered increased chromatin acetylation and thus activation of the silent secondary metabolism *ors* gene cluster in the fungus *Aspergillus nidulans*. Using this model, we aim understanding mechanisms of microbial communication based on bacteria-triggered chromatin modification. Using genome-wide ChIP-seq analysis of acetylated histone H3, we uncovered the unique chromatin landscape in *A. nidulans* upon co-cultivation with *S. rapamycinicus* and relate changes in the acetylation to that in the fungal transcriptome. Differentially acetylated histones were detected in genes involved in secondary metabolism, in amino acid and nitrogen metabolism, in signaling, and encoding transcription factors. Further molecular analyses identified the Myb-like transcription factor BasR as the regulatory node for transduction of the bacterial signal in the fungus and show its function is conserved in other *Aspergillus* species.
DOI: https://doi.org/10.7554/eLife.40969.001

## Introduction

The eukaryotic epigenetic machinery can be influenced by bacteria. For example, bacteria can secrete chromatin modifiers or proteins such as methyltransferases that cause chromatin silencing in eukaryotic cells (*Yoshida et al., 1990*; *Rolando et al., 2013*). As an early example, we discovered that the silent secondary metabolite (SM) gene cluster for orsellinic acid (*ors*) in the filamentous fungus *Aspergillus nidulans* is activated upon physical interaction with the bacterium *Streptomyces rapamycinicus*. The interaction of the fungus with this distinct bacterium led to increased acetylation of histone H3 lysines 9 and 14 at the *ors* gene cluster and thus to its activation (*Schroeckh et al., 2009*; *Nützmann et al., 2011*; *Nützmann et al., 2013*). The lysine acetyltransferase (KAT) responsible for the acetylation and activation of the *ors* gene cluster was shown to be GcnE (*Nützmann et al., 2011*).

Using this model, we aim to gain an understanding of the molecular mechanisms of microbial communication based on bacteria-triggered chromatin modification. In order to obtain a holistic view on the fungal-bacterial interaction that, in the future, might allow predicting interaction partners and discovering the molecular elements involved, we developed a genome-wide chromatin immunoprecipitation (ChIP)-seq analysis specifically during co-cultivation. This led to the discovery of major alterations of epigenetic marks in the fungus triggered by the bacterium and to the identification of BasR as key regulatory node required for linking bacterial signals with the regulation of SM gene clusters.

## Results

### Genome-wide profiles of H3K9 and H3K14 acetylation in *A. nidulans* change upon co-cultivation with *S. rapamycinicus*

*A. nidulans* with and without *S. rapamycinicus* was analyzed by genome-wide ChIP-seq for enrichment of acetylated (ac) histone H3 at lysines K9 and K14 (*Figure 1*; Appendix 1 – Details of ChIP analysis). To account for reads originating from *S. rapamycinicus* we fused the genomes of *A. nidulans* (eight chromosomes) and *S. rapamycinicus*. The resulting fused genome also served as reference for mapping of chromatin marks (see Appendix 1 – Details of the ChIP analysis).

H3K14ac and H3K9ac showed a higher degree of variability across the genome than on H3, implying that the regulatory dynamics of histone acetylation are more specific than those that would be achieved by H3 localization alone. Some areas, such as a region in the first half of chromosome four, were particularly enriched in these acetylation marks, potentially indicating distinctive acetylation islands, which are short loci with continuous enrichment of histone modifications. Such islands have been identified previously in the intergenic and transcribed regions of the human genome, and some of these have been shown to colocalize with known regulatory elements (*Roh et al., 2005*). An island that is particularly enriched for H3K9ac was found around the *ors* gene cluster (*Figure 1c and 2*), thus supporting our previous data (*Nützmann et al., 2011*). The coverage profiles of H3, H3K14ac and H3K9ac consistently change in co-culture compared to monoculture, as seen in *Figure 1*. In particular, the promoter region of the genes *orsD* and *orsA* showed reduced nucleosome occupancy (see *Figure 1c*). This could be due to a redistribution of nucleosomes that ultimately changes the distribution of histone marks. Such nucleosome rearrangements might represent the prevailing driver of H3K14ac change which is associated with a reduction in overall acetylation level. This might explain the local H3K14ac decrease at the translation start sites (TSSs) shown in *Figures 1* and *2*. In comparison to the changes to H3K14ac, the changes to H3K9 acetylation levels are stronger, leading to an increase in H3K9 acetylation despite nucleosome rearrangements. This finding is supported by the observation that unmodified H3 was depleted throughout the *ors* cluster, especially at the *orsA* and *orsD* TSSs.

### Co-cultivation of *A. nidulans* with *S. rapamycinicus* had a major impact on SM gene clusters, nitrogen assimilation, signaling and mitochondrial activity

We employed two strategies to measure changes in histone modification levels. The first analysis was based on the finding that histone acetylation can mostly be found on histones within a gene, in particular on nucleosomes +1 and +2 (*Jiang and Pugh, 2009*) (*Appendix 1—figure 1*). Therefore,

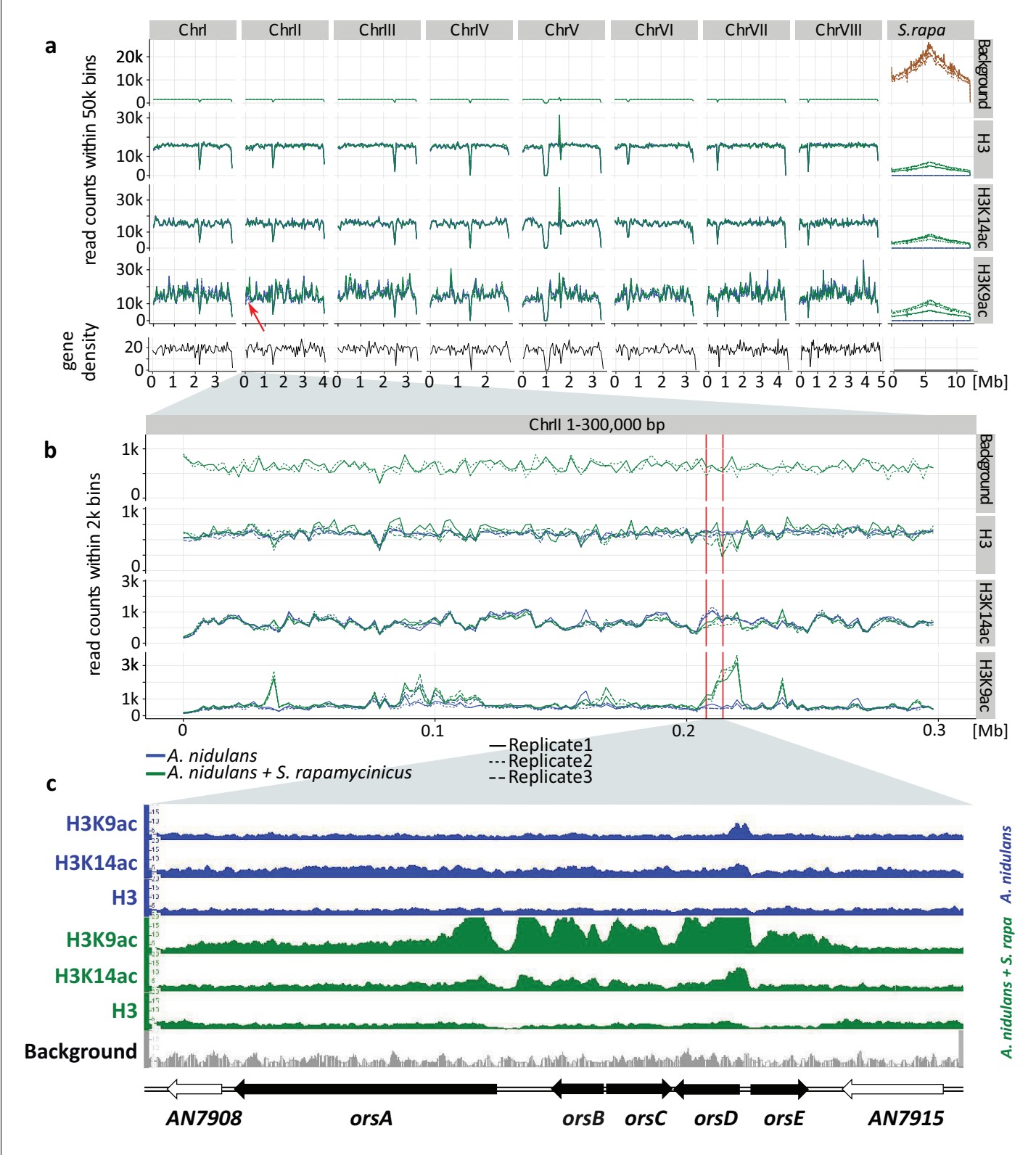

**Figure 1.** Genome-wide coverage plot of the fused fungal-bacterial genome with indication of the C-terminus of H3(Cterm) and acetylated H3 (K9 and K14). For each condition, ChIP-seq analyses of three independent samples were performed. (**a**) Genome-wide analysis covering all chromosomes. Data for all the chromosomes of *A. nidulans* (I to VIII) as well as for the chromosome of *S. rapamycinicus* are shown. The x-axis corresponds to the genome coordinates of the fused genome in Mb. The y-axis corresponds to the number of reads mapping within equally sized

*Figure 1 continued on next page*

*Figure 1 continued*

windows (bins) that segment the fused genome at a resolution of 50 kb for each library separately (see 'Materials and methods' for details). The read count values are plotted at the midpoint of each bin, which are connected by lines. Gene density is reported likewise by counting the number of genes for each bin instead of reads. Background values derive from *S. rapamycinicus* (brown) and *A. nidulans* (green) grown in monoculture. The red arrow indicates the location of the *ors* gene cluster. (b) Zoom into chromosome II. The red lines mark the *ors* gene cluster. Data from three replicates are shown, which share the same tendency. Overall intensities for background, H3K9ac, H3K14ac and H3(Cterm) are compared between *A. nidulans* monoculture (blue) and co-culture (green). The average genome density (black) is also shown. (c) Example of an Integrative Genomics Viewer (IGV) screenshot showing the region of the *ors* gene cluster at the bottom of the figure labeled with black arrows. Other differentially acetylated gene bodies are listed in *Supplementary file 1*. White gene arrows indicate genes that do not belong to the *ors* gene cluster. Data obtained from monocultures of the fungus are depicted in blue and from co-cultivation in green, whereas background data are shown in gray.

DOI: https://doi.org/10.7554/eLife.40969.002

for each library, we counted mapped reads that overlapped genes. This formed the basis for a quantitative comparison between monocultures and co-cultures using standard read-counting methods for sequencing data (see 'Materials and methods'). Throughout this study, we refer to this method as differential chromatin state (DCS) analysis. The second analysis was based on a first round of peak-calling and subsequent quantification of the peaks. Comparison of the generated data sets showed 84 ± 1.7% similarity. The data obtained from the gene-based DCS method (*Supplementary file 1*) were used for both further analyses and comparisons of the culture conditions using a false discovery rate (FDR) cut-off of 0.01. This does not include further filtering on the log-fold changes (LFCs) to capture the possible biological relevance of the detected changes.

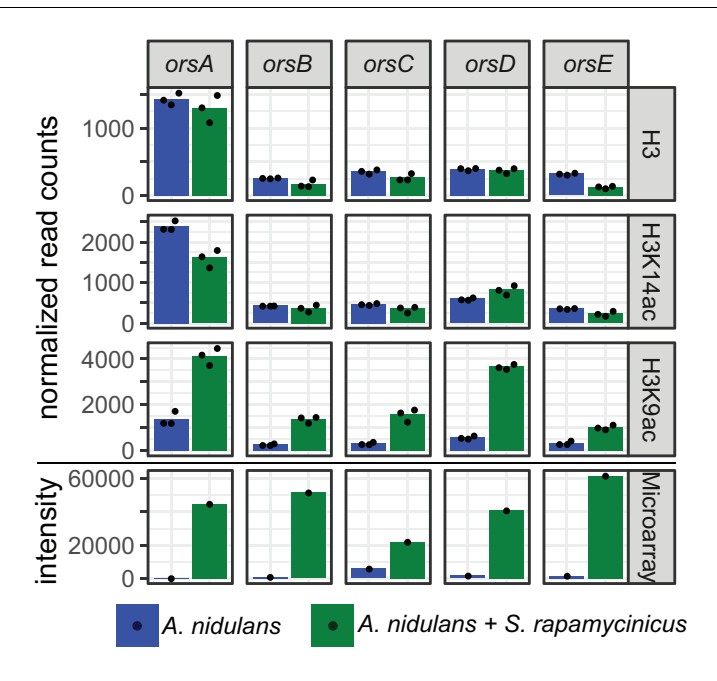

**Figure 2.** Normalized read counts derived from differential chromatin state (DCS) analysis obtained for the *ors* genes based on H3, H3K14ac and H3K9ac ChIP-seq. Data were generated for the area 500 bp down- and 1000 bp upstream of the TSSs. Depicted bars are calculated from three data points.

DOI: https://doi.org/10.7554/eLife.40969.003

The following figure supplements are available for figure 2:

**Figure supplement 1.** Normalized ChIP-seq read counts were used to quantify the chromatin state of individual genes.

DOI: https://doi.org/10.7554/eLife.40969.004

**Figure supplement 2.** Relation between ChIP-seq and microarray data.

DOI: https://doi.org/10.7554/eLife.40969.005

Quality and the absence of possible biases introduced by the co-culture or other sources were further investigated by MA plots. They showed a symmetrical and even distribution around LFC = 0, meeting the requirements for the statistical tests described in the 'Materials and methods' (*Appendix 1—figure 2*). DCS analysis of H3, as a proxy for nucleosome occupancy, was found to be lower (FDR < 0.01) in 37 genes and higher in two genes during co-cultivation. Using the same cut-off, H3K14ac levels during bacterial-fungal co-cultivation were found to be lower for 154 genes and higher for 104 genes. Differential acetylation of chromatin was found for H3K9ac, with 297 genes with significantly lower and 593 with significantly higher acetylation (*Supplementary file 1*).

The analysis of microarray data obtained under identical conditions showed a positive correlation of higher gene expression with H3K9 acetylation (r = 0.2 for all genes and r = 0.5 for a subset of genes showing differential acetylation; *Appendix 1—figures 3* and *4*). Data for selected genes are summarized in *Supplementary file 2*, which shows the LFCs of H3K9ac ChIP-seq data with their corresponding microarray data. In total, higher acetylation during co-cultivation was seen in histones belonging to six SM gene clusters, the *ors*, aspercryptin (*atn*), cichorine (*cic*), sterigmatocystin (*stc*), anthrone (*mdp*) and 2,4-dihydroxy-3-methyl-6-(2-oxopropyl)benzaldehyde (*dba*) gene clusters, with the emericellamide (*eas*) and microperfuranone clusters being the only ones with reduced acetylation and expression (*Supplementary file 2*, section V). With a few exceptions, the genes covered by histone H3 that had increased acetylation are involved in calcium signaling and asexual development

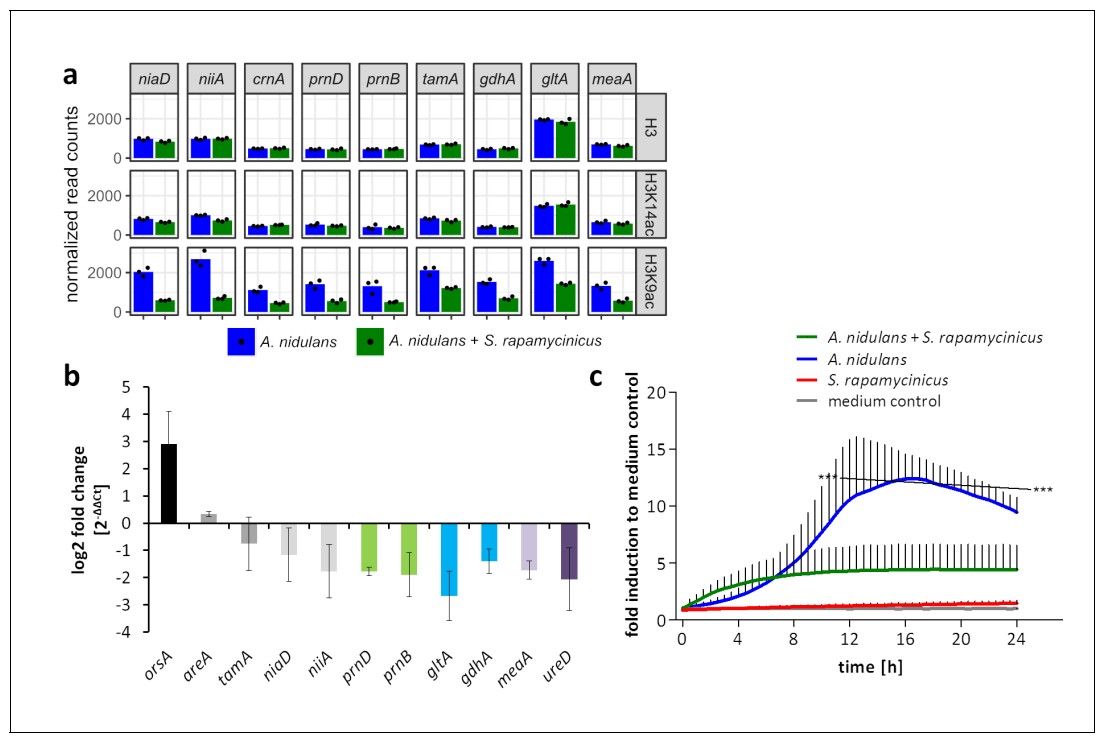

**Figure 3.** Influence of *S. rapamycinicus* on fungal nitrogen metabolism and mitochondrial functions. (a) Normalized ChIP-seq read counts were used to quantify the chromatin state (H3, H3K14ac, H3K9ac) of nitrogen metabolism genes. Counts were obtained by counting reads mapping to the promoter area of each gene, which is defined as the sequence 500 bp down- and 1000 bp upstream from the TSSs. Depicted bars are calculated from three data points. (b) Transcription analysis of randomly selected genes of primary and secondary nitrogen metabolism by qRT-PCR during co-cultivation. Relative mRNA levels were measured after 3 hr and normalized to the β-actin gene expression. The transcription of *orsA* was used as a positive control. (c) Respiratory activity comparing *A. nidulans* grown in co-culture with *S. rapamycinicus* and *A. nidulans* in monoculture. Respiratory activity was determined using a resazurin assay. Data were normalized to medium. The black line shows the time points that are significantly different between *A. nidulans* and *A. nidulans* grown in co-culture with *S. rapamycinicus*. ***p<0.001.

DOI: https://doi.org/10.7554/eLife.40969.006

The following figure supplement is available for figure 3:

**Figure supplement 1.** Gene ontology of the 15 most significantly enriched categories for differentially higher and lower acetylated genes at H3K9 upon co-cultivation with *S. rapamycinicus*.

DOI: https://doi.org/10.7554/eLife.40969.007

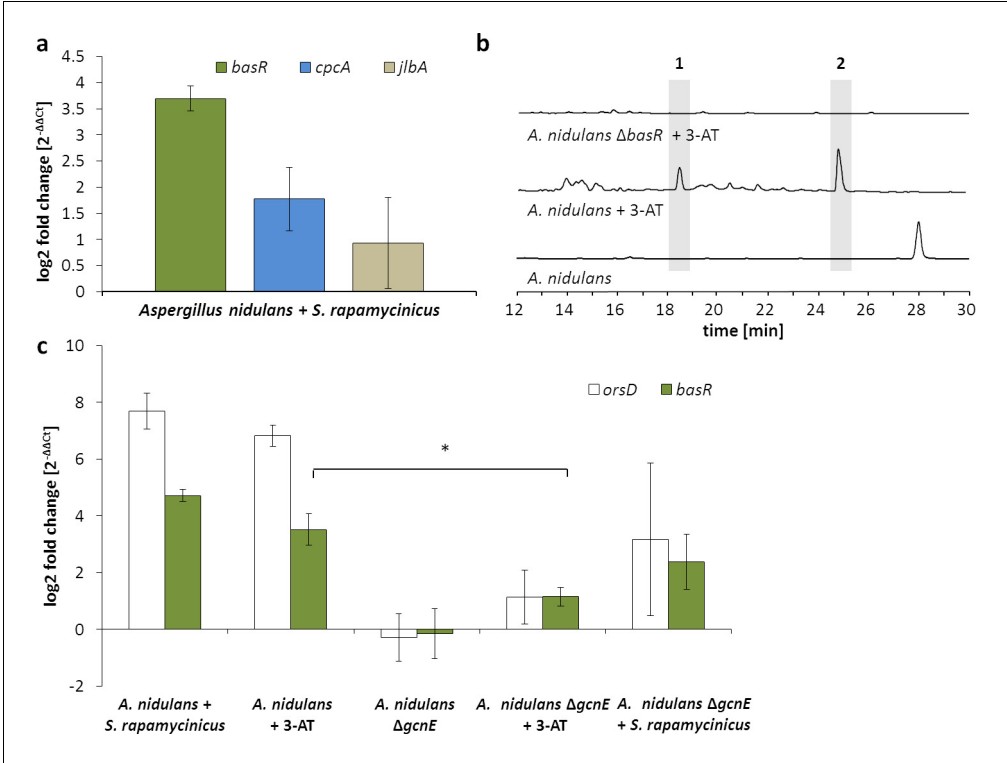

**Figure 4.** Artificial histidine starvation using 3-AT led to *ors* gene cluster activation. (**a**) Transcription of *basR, cpcA* and *jlbA* determined by qRT-PCR after 3 hr of co-cultivation. Relative mRNA levels were compared to β-actin gene expression. (**b**) High-performance liquid chromatography (HPLC)-based detection of orsellinic acid (1) and lecanoric acid (2) in supernatants of *A. nidulans* cultures treated with 3-AT. (**c**) Relative transcript levels of *orsA, cpcA* and *basR* 6 hr after 3-AT addition to the *A. nidulans* monoculture and the *gcnE* deletion mutant. *$p < 0.05$.
DOI: https://doi.org/10.7554/eLife.40969.008

(*Supplementary file 2*, sections III and IV; *Figure 2—figure supplement 1*). A major group of genes with reduced acetylation in mixed cultivation compared to the monoculture of *A. nidulans* is linked to the fungal nitrogen metabolism (*Supplementary file 2*, section I) including genes for the utilization of primary and secondary nitrogen sources, such as genes of the nitrate assimilation gene cluster and the glutamine dehydrogenase gene (*Figure 3a and  – Figure 3—figure supplement 1*). These data were confirmed by quantifying the expression of identified genes by qRT-PCR (*Figure 3b*).

Genes assigned to mitochondrial function showed decreased acetylation of H3K9, which implied reduced mitochondrial function. This assumption was confirmed by measuring the respiratory activity of fungal cells. In monoculture, the fungus showed a high metabolic activity, which was significantly reduced during co-cultivation (*Figure 3c*).

## Bacteria induce elements of the fungal cross-pathway control

To identify transcription factors that are involved in transducing the bacterial signal to the fungal expression machinery, and because a transcription factor gene is missing in the *ors* gene cluster, we searched the 890 differentially H3K9 acetylated genes for those annotated as putatively involved in transcriptional regulation. In total, 22 putative transcription factor-encoding genes fulfilled this requirement (*Supplementary file 2*, section VII). Most of them (18 genes) showed significantly higher acetylation in co-culture, whereas only four genes had lower acetylation. Among the genes with increased acetylation in co-culture were *cpcA*, coding for the central transcriptional activator of the cross-pathway control CpcA, as well as the bZIP transcription factor gene *jlbA* (jun-like bZIP). Both of these genes have been shown to be highly expressed during amino-acid starvation in *A. nidulans* (*Hoffmann et al., 2001*; *Strittmatter et al., 2001*). In addition, a putative ortholog (AN7174) of the

*S. cerevisiae bas1* gene showed an increase in acetylation. In yeast, Bas1p (together with the homeo-domain protein Bas2p) is involved in the regulation of amino-acid biosynthesis (*Springer et al., 1996*; *Valerius et al., 2003*). Consistently, a number of genes related to amino-acid metabolism showed increased acetylation of H3K9 during the co-cultivation of *A. nidulans* with *S. rapamycinicus* (*Supplementary file 2*, section II). qRT-PCR analysis was carried out to correlate the ChIP-seq data with the expression levels of *cpcA, jlbA* and *AN7174* , and this demonstrated upregulation of *cpcA* and *AN7174* during co-cultivation (*Figure 4a*). In *S. cerevisiae*, it was shown that Gcn4 (CpcA in *A. nidulans*) and Bas1p share a similar DNA-binding motif and that both activate the transcription of the histidine biosynthesis gene *HIS7* independently of each other (*Springer et al., 1996*). Consistent with a possible involvement of these transcription factors in cross-pathway control (CPC) is the observation that the addition of the histidine analogue 3-aminotriazole (3-AT), which is known to induce the CPC via amino-acid starvation, led to the production of orsellinic acid in the fungal monoculture (*Figure 4b*) and to an increased expression of *orsA, cpcA* and *AN7174* (*Figure 4c*).

To analyze a possible involvement of these genes in the bacteria-induced activation of the *ors* gene cluster, the genes *cpcA* (data not shown) and *AN7174* (*Figure 5—figure supplement 1a*) were deleted. Deletion of *cpcA* in *A. nidulans* showed no effect on the induction of the *ors* gene cluster in response to *S. rapamycinicus* (data not shown), whereas deletion of *AN7174* resulted in a significantly reduced expression of *orsA* and *orsD*, and in complete loss of orsellinic acid production (*Figure 5*). Therefore, *AN7174* was named *basR* and analyzed in detail.

## The transcription factor BasR is a central regulatory node in bacteria-triggered regulation of the SM gene cluster

Further analysis of the *A. nidulans* genome revealed a second gene (*AN8377*) encoding a putative ortholog of the *S. cerevisiae bas1* gene (*Figure 6* and *Figure 6—figure supplement 1*). Both genes (*basR* and *AN8377*) code for Myb-like transcription factors whose function in filamentous fungi is completely unknown. We compared the H3K9 acetylation and gene expression of both genes upon co-cultivation. The *basR* gene showed increased H3K9 acetylation (LFC = 0.6) and drastically increased transcription (LFC = 5.85) during co-cultivation compared to *AN8377* (H3K9ac

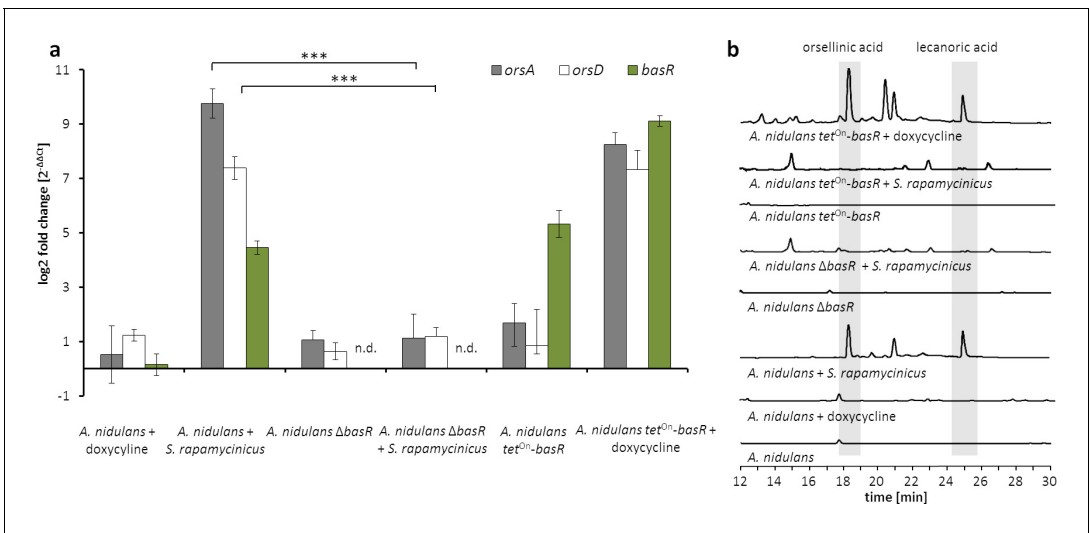

**Figure 5.** The Myb-like transcription factor BasR of *A. nidulans* is required for the activation of the *ors* gene cluster. (**a**) Relative transcript levels of *ors* cluster genes *orsA, orsD* and *basR* after 6 hr of cultivation in Δ*basR* mutant strain and *tet*$^{On}$-*basR* overexpression strain incubated with and without doxycycline. Transcript levels were measured by qRT-PCR normalized to β-actin transcript levels. (**b**) HPLC-based detection of orsellinic and lecanoric acid in the wild-type strain, *basR* deletion mutant and *basR* overexpression strain. n.d.: not detectable; ***p<0.001.

DOI: https://doi.org/10.7554/eLife.40969.009

The following figure supplement is available for figure 5:

**Figure supplement 1.** Generation of a *basR* deletion mutant and an inducible overexpression strain based on the *A. nidulans* wild-type strain A1153.

DOI: https://doi.org/10.7554/eLife.40969.010

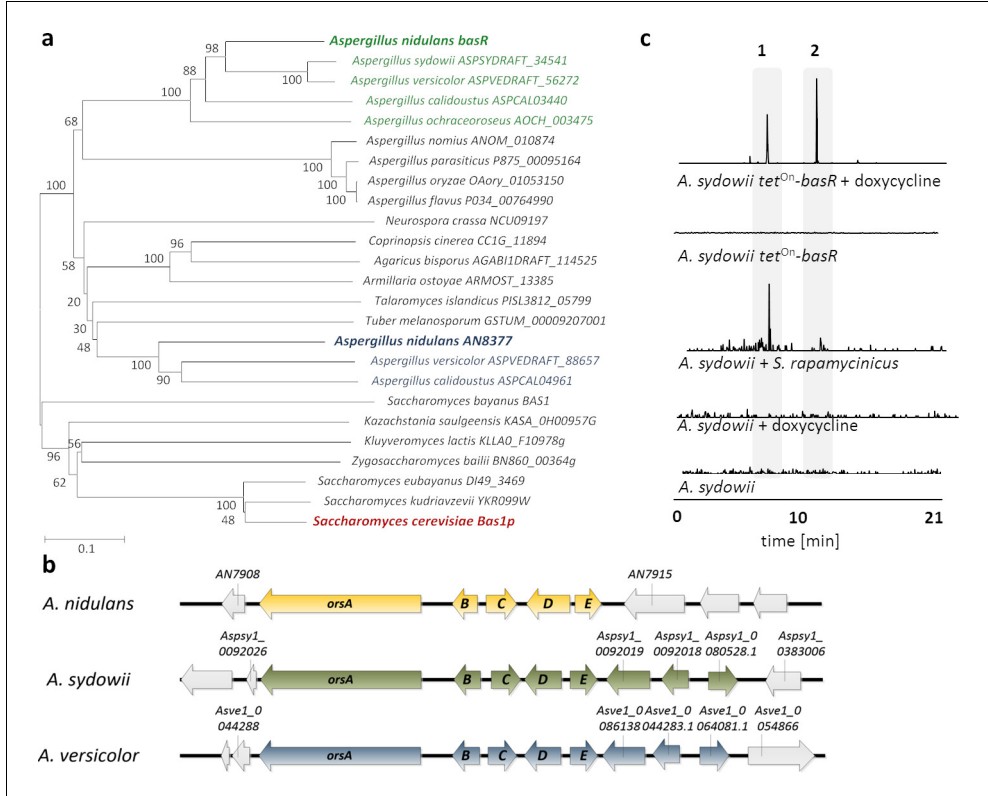

**Figure 6.** Co-occurrence of BasR and the orsellinic acid gene cluster in other fungi is linked to the *S. rapamycinicus*-triggered *ors* gene cluster activation. (**a**) Phylogenetic analysis of BasR (*AN7174; green*) showing its position among other fungi. The percentage of trees in which the associated taxa clustered together is shown next to the branches. The names of the selected sequences are given according to their UniProt accession numbers. A comprehensive phylogenetic tree is depicted in *Figure 6—figure supplement 1*. (**b**) Alignment of the orsellinic acid gene clusters in the fungal species containing a *basR* homologue (*A. nidulans*, *A. sydowii* and *A. versicolor*), where *orsA* encodes the polyketide synthase, whereas *orsB-orsE* code for tailoring enzymes. (**c**) Liquid chromatography–mass spectrometry (LC-MS)-based detection of orsellinic and lecanoric acid in monoculture of the *A. sydowii basR* overexpression strain following induction with doxycycline and during co-cultivation of *A. sydowii* and *S. rapamycinicus*. LC-MS profiles of the extracted ion chromatogram (EIC) are shown for *m/z* 167 [M – H]⁻, which corresponds to orsellinate. Orsellinic (1) and lecanoric acid (2) were detected via their fragment ion orsellinate.

DOI: https://doi.org/10.7554/eLife.40969.011

The following figure supplements are available for figure 6:

**Figure supplement 1.** Molecular phylogenetic analysis of BasR (*AN7174*).

DOI: https://doi.org/10.7554/eLife.40969.012

**Figure supplement 2.** Deletion of the second putative *bas1p* homologous gene (*AN8377*) in *A. nidulans* and analysis of its impact on the *ors* gene cluster induction in response to *S. rapamycinicus*.

DOI: https://doi.org/10.7554/eLife.40969.013

**Figure supplement 3.** Generation of the inducible *basR*-overexpression strain by ectopic integration of an additional copy of the *basR* gene in the *A. sydowii* wild type strain (wt).

DOI: https://doi.org/10.7554/eLife.40969.014

LFC = −0.03; microarray LFC = 0.14). Deletion of *AN8377* (*Figure 6—figure supplement 2a*) did not affect the induction of fungal orsellinic acid production upon co-cultivation (*Figure 6—figure supplement 2b*), excluding a role for *AN8377* in this process.

In *S. cerevisiae*, Bas1p needs the interaction with Bas2p for the transcriptional activation of several genes that are required for histidine and purine biosynthesis (*Springer et al., 1996*). The C-terminal activation and regulatory (BIRD) domain of Bas1, which was described as mediating this Bas1p–Bas2p interaction (*Pinson et al., 2000*), is missing in BasR. It is thus not surprising that we did

not find an ortholog for the *S. cerevisiae bas2* gene in the *A. nidulans* genome. Although the addition of 3-AT to monocultures of *A. nidulans* led to the production of orsellinic acid and derivatives thereof, the effect of 3-AT was abolished in the *basR* deletion mutant strain (*Figure 4b*).

The transcriptional activation of *HIS7* by Bas1/Bas2 upon adenine limitation in yeast requires a functional Gcn5 (GcnE in *A. nidulans*) (*Valerius et al., 2003*), so we raised the question of whether GcnE is needed for full *basR* expression. Addition of *S. rapamycinicus* or 3-AT to the *gcnE* deletion mutant led to decreased *basR* gene expression compared to levels of gene expression seen in the wild type in co-culture or in a monoculture with 3-AT (*Figure 4c*). These data indicate that GcnE is required for *basR* expression. Inspection of the *basR* mutant strain on agar plates did not reveal further obvious phenotypes (data not shown).

To further substantiate the influence of *basR* on the *ors* gene cluster, we generated a *basR* over-expression strain (*Figure 5—figure supplement 1b*) by employing the inducible *tet*^On^-system (*Helmschrott et al., 2013*). Addition of doxycyline to the media induced *basR* expression as well as the expression of the *ors* gene cluster (*Figure 5a*). However, *basR* gene expression was detectable even without doxycycline addition, indicating 'leakiness' of the *tet*^On^-system. Nevertheless, production of orsellinic and lecanoric acid was only detected upon doxycycline addition (*Figure 5b*), supporting the important role of BasR for their biosynthesis. To address the question of whether other SM biosynthesis gene clusters are regulated by BasR, we performed RNA-sequencing (RNA-seq) analysis. We examined the transcription profiles of *A. nidulans* with and without *S. rapamycinicus*, and compared it to those of the *basR* overexpression strain with and without the addition of doxycycline. Obvious candidate gene clusters, regulated by BasR, were the eight differentially acetylated gene clusters found in the ChIP-seq data (*Supplementary file 2*). Five of the eight differentially acetylated SM gene clusters, namely the *dba*, *cic*, *eas* and microperfuranone gene clusters, were also differentially transcribed in response to the streptomycete as well as in the *basR* overexpression strain (*Figure 7a*; *Supplementary file 3*), emphasizing the importance of BasR in bacteria-induced secondary metabolite regulation. In addition, this finding was perfectly mirrored when we applied matrix assisted laser desorption/ionization (MALDI)-mass spectrometry (MS) imaging, which showed reduced levels of emericellamides both in *basR*-overproducing colonies of *A. nidulans* and in co-grown colonies, but not in colonies without the streptomycete or doxycycline addition (*Figure 7b*).

Interestingly the microperfuranone gene cluster, which is acetylated at lower levels and transcribed in response to the bacterium, is transcriptionally upregulated in the *basR*-overexpression strain, suggesting a transcriptional regulation that is independent of the signal(s) induced by *S. rapamycinicus*.

## The presence of BasR in fungal species makes it possible to forcast the inducibility of *ors*-like gene clusters by *S. rapamycinicus*

To address the question of whether *basR* homologs exist in other fungi and whether such potential homologs have similar functions, we analyzed fungal genomes using BlastP. Surprisingly, obvious *basR* homologs are only present in a few other *Aspergillus* spp. including *Aspergillus sydowii* and *Aspergillus versicolor*, and are apparently lacking in many others (*Figure 6 and Figure 6—figure supplement 1*). Interestingly, in addition to three additional genes in both fungi, a gene cluster similar to the *ors* gene cluster of *A. nidulans* was also identified (*Figure 6b*). We overexpressed *basR* in *A. sydowii* using the *tet*^On^-system to analyze its function (*Figure 6—figure supplement 3*). LC-MS analyses revealed the appearance of novel masses that were assigned to orsellinic acid derivatives (*Figure 6c*).

Finally, we addressed the question of whether the presence of the *basR* gene and the *ors* gene cluster allows the forecasting of their inducibility by *S. rapamycinicus*. As shown in *Figure 6*, also co-cultivation of *A. sydowii* with *S. rapamcinicus* led to the activation of the fungal *ors* gene cluster, again linking BasR with the bacteria-triggered induction of the production of orsellinic acid derivatives.

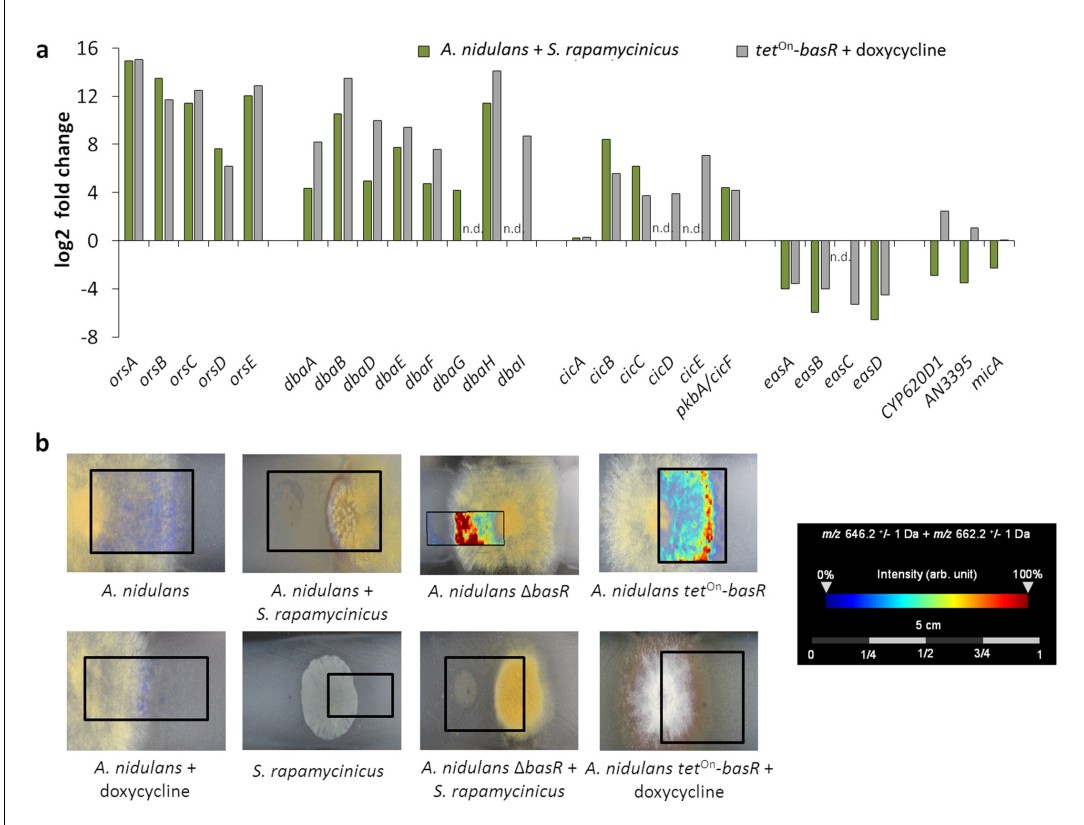

**Figure 7.** The Myb-like transcription factor BasR of *A. nidulans* is required for *S. rapamycinicus*-triggered regulation of SMs. (a) Transcript levels of the *ors*, *dba*, cichorine, *eas* and microperfuranone gene clusters in *A. nidulans* co-cultivated with *S. rapamycinicus* and in the *basR* overexpression mutant treated with doxycycline to induce *basR* gene expression. Transcripts per million (TPM) values were divided by values for *A. nidulans* monoculture and the untreated *basR*-overexpression strain to obtain fold changes. (b) Visualization of ions *m/z* 646.3 and *m/z* 662.3 ± 1 Da, potentially corresponding to [M + Na]$^+$ and [M + K]$^+$ of emericellamide E/F (C$_{32}$H$_{57}$N$_5$O$_7$; accurate mass 623.4258), by MALDI-MS imaging. Images were corrected by median normalization and weak denoising. n.d.: not detectable.

DOI: https://doi.org/10.7554/eLife.40969.015

## Discussion

### *S. rapamycinicus* induces a unique chromatin landscape in *A. nidulans*

We were able to use genome-wide ChIP-seq analysis of acetylated histone H3 (H3K9ac, H3K14ac) and the quantification of H3 to uncover the chromatin landscape in the fungus *A. nidulans* upon co-cultivation with *S. rapamycinicus*. In an attempt to characterize the general distribution of nucleosomes and acetylation marks over the genome, we compared the intensity of chromatin states with gene density. A lower gene density was typically found in heterochromatic regions such as the centromeres and telomeres, creating a repressing environment (*Allshire and Ekwall, 2015*). We found reduced H3 occupancy in heterochromatic regions, indicating either replacement of H3 by the centromere-specific H3 CENP-A or reduced nucleosome occupancy (*Smith et al., 2011*; *Allshire and Ekwall, 2015*).

We observed distinct peaks for H3K9ac in *A. nidulans* grown in co-culture with *S. rapamycinicus*. One of the areas with the greatest increase in H3K9ac was the *ors* gene cluster, nicely confirming our previous findings (*Nützmann et al., 2011*) (*Figure 8*). Furthermore, previous ChIP qRT-PCR experiments indicated a distinct increase of H3K9ac inside the cluster borders, which did not expand to neighboring genes (*Nützmann et al., 2011*). By contrast, the H3K14ac modification seemed to be of a more global nature and not exclusively confined to specific regions such as the *ors* gene cluster. These conclusions were extended here by the pattern detected in the genome-wide ChIP-seq data, which showed no spreading of H3K9ac to genes adjacent to the *ors* gene cluster,

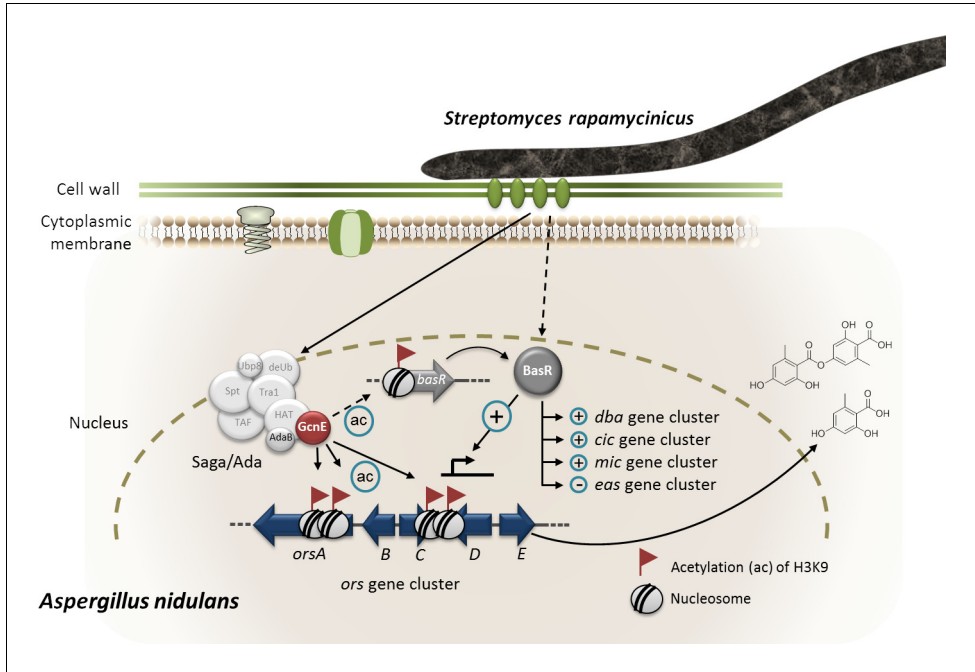

**Figure 8.** Model of *S. rapamycinicus – A. nidulans* interaction. Co-cultivation leads to activation of the *basR* gene. The lysine acetyltransferase GcnE specifically acetylates (ac) lysine (K)9 of histone H3 at the *ors* gene cluster and presumably at the *basR* gene promoter. As a consequence, *basR* is expressed. The transcription factor BasR activates (+) and represses (–) the expression of the *ors*, *cic,* microperfuranone (*mic*) and *eas* gene clusters directly or indirectly. The involvement of AdaB and GcnE of the Saga/Ada complex has been experimentally proven (***Nützmann et al., 2011***).

DOI: https://doi.org/10.7554/eLife.40969.016

thereby demonstrating the quality of the genome-wide ChIP data generated here. Furthermore, these results are also consistent with our previous finding of reduced expression of SM cluster genes as a consequence of the lack of H3K14 acetylation (***Nützmann et al., 2013***). In contrast to H3K14ac, H3K9ac is less uniformly distributed over the genome. It only showed strong enrichment in the promoters of certain genes. Especially high levels of acetylation were found at *orsA* and in the bidirectional promoters of *orsD* and *orsE*. This observation was recently confirmed by the finding that H3ac and H3K4me3 were increased at the *orsD* gene only when the *ors* cluster was transcriptionally active (***Gacek-Matthews et al., 2016***).

We also assessed the distribution of H3K9ac and H3K14ac, as well as that of the C-terminus of H3 (H3Cterm), at the TSSs and translation termination sites (TTSs) (Appendix 1 – Chromatin profiles at translation start sites and translation termination sites). For H3K9, an enrichment of acetylation ~500 bp downstream of the TSSs as well as immediately upstream of the TSSs was observed. This was expected as similar results were obtained with an antibody targeting the acetylated N-terminus of histone H3 in *A. nidulans* (***Gacek-Matthews et al., 2016***) and in other fungi such as *S. cerevisiae* and *Cryptococcus neoformans* (***Haynes et al., 2011***; ***Mews et al., 2014***). Increased acetylation coincides with reduced levels of H3 around the TSSs, which are most probably due to a depletion of nucleosomes at the promoter. The profile plots for H3K14 acetylation are similar, although not as highly enriched around the TSSs as those for H3K9 (***Appendix 1—figure 5 a***). As expected, a comparison of LFCs for both modifications showed high similarity, suggesting that the modifcations are established interdependently (***Gacek and Strauss, 2012***; ***Waters et al., 2015***). At the 3' end of the ORF, H3 density drastically increased accompanied by reduced levels of H3K9ac and H3K14ac (***Appendix 1—figure 5***). Likewise, reduced acetylation at the TTSs was observed in *A. nidulans* (***Gacek-Matthews et al., 2016***) and *S. cerevisiae* (***Mews et al., 2014***). It is interesting to notice that the increase in nucleosome density directly correlated with a decrease in the gene expression rate

(*Appendix 1—figure 6e*). Previous studies suggested a direct correlation between the presence of nucleosomes and the stalling of RNA polymerase II (*Grosso et al., 2012*).

## Increased gene expression directly correlates with histone H3K9 acetylation

Acetylation is generally regarded as an activating chromatin mark that promotes the transcription of eukaryotic genes (*Bannister and Kouzarides, 2011*). Our study suggests a more differentiated picture. When we compared data from this study with microarray data (*Nützmann et al., 2011*) (*Appendix 1—figures 3* and *4*), the acetylation of H3K9 directly correlated with gene expression levels. A similar finding was reported for other fungi (*Wiemann et al., 2013*). By contrast, this was not observed for the acetylation of H3K14. This could partly result from the low number of targets for this modification. By contrast, gene promoters showed a distinct increase of H3K14ac at the TSSs in dependence on the average transcription level (*Appendix 1—figure 6c*). The low correlation between active gene transcription and acetylation at H3K14 confirmed earlier results (*Reyes-Dominguez et al., 2008*; *Nützmann et al., 2011*). Previously, we showed that a mimicry of a hypo-acetylated lysine 14 on histone H3 drastically altered the phenotype and the expression of SM gene clusters (*Nützmann et al., 2013*). This effect was overcome, however, when later time points of cultivation were considered. Taken together, the primary location at the TSSs and the major defect in SM production at earlier stages indicate a role for H3K14ac in transcriptional initiation. Hyper-acetylation at H3K14 could be also relevant for marking active genes and providing a docking site for regulatory proteins.

## *S. rapamycinicus* silences fungal nitrogen metabolism

A substantial number of genes that are involved in primary and secondary nitrogen metabolism were strongly depleted for H3K9ac upon co-cultivation with *S. rapamycinicus*. This correlated with reduced expression of the respective genes. Thus, upon contact with the bacterium, *A. nidulans* showed reduced nitrogen uptake and reduced degradation of various nitrogen sources, leading to nitrogen starvation.

Under nitrogen starvation or low availability of primary nitrogen sources, such as glutamine and ammonium, the intracellular level of glutamine drops (*Tudzynski, 2014*). This was in fact observed for the intracellular concentration of amino acids in *A. nidulans* when the fungus was co-cultured with the bacterium (*Appendix—figure 7*). Thus, in presence of *S. rapamycinicus* but not of non-inducing streptomycetes such as *S. lividans* the fungus is in a physiological state of nitrogen starvation (*Figure 8*). Nitrogen limitation has been shown before to represent a trigger for the activation of a number of SM gene clusters including the *ors* gene cluster (*Scherlach et al., 2011*; *Studt et al., 2012*). Nitrogen starvation also activates the expression of the anthrone (*mdp*) gene cluster (*Scherlach et al., 2011*), which we also observed in our data. However, induction of orsellinic acid production by nitrogen starvation took about 60 hr, whereas co-cultivation with *S. rapamycinicus* had already triggered expression of the cluster genes after 3 hr. Therefore, it is unlikely that the bacteria-triggered activation of the cluster is exclusively achieved by restricting nitrogen availability for the fungus. Furthermore, shortage of nitrogen leads to de-repression of genes that are involved in the usage of secondary nitrogen sources, which was not supported by our data. In *S. cerevisiae*, it has been reported that a shift from growth under nutrient sufficiency to nitrogen starvation induced the degradation of mitochondria (*Eiyama et al., 2013*). Similarly, decreased acetylation and transcription of genes with mitochondrial function were also detected upon contact of *A. nidulans* with the bacterium. This was further supported by a lower mitochondrial respiratory activity in the fungal cells during co-cultivation (*Figure 3c*).

## BasR is a central regulatory node for integrating bacterial signals leading to regulation of SM gene clusters

Another consequence of nitrogen starvation is the reduced availability of amino acids in the cell. Consequently, as shown here, the amino-acid biosynthetic pathways represented a major group of de-regulated genes at both the acetylation and expression levels. Amino-acid biosyntheses in fungi are regulated by the CPC system upon starvation for distinct amino acids (*Tudzynski, 2014*; *Krappmann and Braus, 2005*). Since deletion of *cpcA* in *A. nidulans* did not affect the induction of

the *ors* gene cluster, whereas the artificial inducer of the CPC system 3-AT does (*Sachs, 1996*), it is conceivable that CPC somehow plays a role. 3-AT is a structural analogue of histidine that triggers histidine starvation in the fungal cell and thereby the CPC (*Sachs, 1996*). In *S. cerevisiae*, other regulators such as the heterodimeric transcription factor complex Bas1p/Bas2p, which is even bound by Gcn5p, have also been shown to induce the CPC (*Valerius et al., 2003*; *Daignan-Fornier and Fink, 1992*). We identified two putative orthologous genes in the genome of *A. nidulans*, but further analysis revealed that only *basR* (*AN7174*) was involved in *ors* gene cluster activation during the fungal-bacterial co-cultivation (*Figure 8*). Despite the fact that *AN8377* seems to resemble *S. cerevisiae bas1* more closely (*Figure 7* and *Figure 6—figure supplement 1*), it is not needed for the *ors* gene cluster activation.

On the basis of bioinformatic analysis, BasR of *A. nidulans* consists of 305 amino acids and thus is rather different from its closest homolog which is Bas1p of *S. cerevisiae* with 811 amino acids (*Zhang et al., 1997*). The BIRD region of Bas1p that mediates the Bas1p-Bas2p interaction (*Pinson et al., 2000*) is missing in BasR. The *basR* gene was highly upregulated in the microarray data, and the upregulation of this gene coincided with the increased H3K9 acetylation of its promoter. *basR* deletion and overexpression clearly demonstrated the function of this transcription factor gene in activating the *ors* gene cluster in response to *S. rapamycinicus*. A functional GcnE seems to be required for efficient *basR* expression, indicating a dependency similar to that observed for *bas1* in yeast (*Valerius et al., 2003*).

Interestingly, the *basR* gene could not be found in all of the fungal genomes analyzed here but it was found, for example, in *A. sydowii* and *A. versicolor*, which were also found to encode *ors* gene clusters. As in *A. nidulans*, overexpression of the *A. sydowii basR* gene led to the activation of its silent *ors* gene cluster. On the basis of this finding, we predicted that *S. rapamycinicus* also induces the *ors* gene cluster in *A. sydowii* an this was indeed the case. We did not find a *basR* homolog in *A. fumigatus*, although the formation of fumicyclines is induced by *S. rapamycinicus* (*König et al., 2013*). This might be due to the fact that the available genome data lack the *basR* gene due to missing annotation or, alternatively, because a different regulatory response mechanism to *S. rapamycinicus* is present in *A. fumigatus*.

Genome-wide ChIP-seq analysis also indicated that the interaction of *S. rapamycinicus* with *A. nidulans* influenced other SM gene clusters and leads to a downregulation of the fungal nitrogen metabolism (Figure 3 and *Supplementary file 2*), which might be regulated via BasR. Further analyses revealed that BasR is also required for the transcriptional regulation of the *dba, cic,* microperfuranone and *eas* gene clusters (*Figure 7*), as well as being important for the downregulation of genes belonging to the nitrate-assimilation gene cluster (*Supplementary file 3*). These data indicate that overexpression of *basR* phenocopies the regulation by *S. rapamycinicus* and highlights the importance of BasR for the regulation of SM gene clusters and its role in transducing the bacterial signal(s) in the fungus. As implied by the finding that the presence of *basR* and the *ors* cluster in several fungi coincided with their inducibility by *S. rapamycinicus*, in future it might be possible to predict which microorganisms communicate with each other based on their genetic inventory.

## Materials and methods

**Key resources table**

| Reagent type (species) or resource | Designation | Source or reference | Identifiers | Additional information |
|---|---|---|---|---|
| Strain, strain background (*Aspergillus nidulans*) | FGSC A1153 | *Nayak et al., 2006* | | *yA1, pabaA1; argB2; pyroA4, nkuA::bar* |
| Strain, strain background (*Aspergillus nidulans*) | A1153ΔgcnE | *Nützmann et al. (2011)* | | *yA1, pabaA1; gcnE::argB2; pyroA4, nkuA::bar* |
| Strain, strain background (*Aspergillus nidulans*) | A1153ΔbasR | This study | | *yA1, pabaA1; basR::argB2; pyroA4, nkuA::bar* |

*Continued on next page*

*Continued*

| Reagent type (species) or resource | Designation | Source or reference | Identifiers | Additional information |
|---|---|---|---|---|
| Strain, strain background (*Aspergillus nidulans*) | A1153tet<sup>On</sup>-basR | This study | | yA1, pabaA1; argB2::pabaA1-tet<sup>On</sup>-basR; pyroA4, nkuA::bar |
| Strain, strain background (*Aspergillus nidulans*) | A1153ΔAN8377 | This study | | yA1, pabaA1; AN8377::argB2; pyroA4, nkuA::bar |
| Strain, strain background (*Aspergillus nidulans*) | A1153gcnE-3xflag | *Nützmann et al., 2011* | | yA1, pabaA1; gcnE::gcnEp-gcnE-3x-flag-pabaA1; pyroA4, nku::bar |
| Strain, strain background (*Aspergillus sydowii*) | CBS 593.65 | Westerdijk Fungal Bio Diversity Institute, The Netherlands | | |
| Strain, strain background (*Aspergillus sydowii*) | A. sydowii tet<sup>On</sup>-basR | This study | | Ectopic integration of pUC18 tet<sup>ON</sup>-A. sydowii basR-hph |
| Strain, strain background (*Streptomyces rapamycinicus*) | ATCC 29253 | *Kumar and Goodfellow, 2008* | | |
| Strain, strain background (*Streptomyces lividans*) | TK24 | *Cruz-Morales et al., 2013* | | |
| Antibody | ANTIFLAG M2 | Sigma-Aldrich, Taufkirchen, Germany | F3165-5MG | |
| Antibody | Rabbit polyclonal anti-histone H3 | Abcam, Cambridge, UK | ab1791 | |
| Antibody | Rabbit polyclonal histone H3K9ac | Active Motif, La Hulpe, Belgium | 39137 | |
| Antibody | Rabbit polyclonal anti-acetyl-histone H3 (Lys14) | Merck Millipore, Darmstadt, Germany | 07 – 353 | |
| Commercial assay or kit | Universal RNA Purification Kit | Roboklon, Berlin, Germany | E3598 | |
| Chemical compound, drug | Digoxigenin-11-dUTP | Jena BioScience, Jena, Germany | NU-803 | |
| Software, algorithm | GraphPad Prism 5 | GraphPad Software Inc., La Jolla, USA | | |
| Software, algorithm | Bioconductor package regioneR | *Gel et al. (2016)* | | |
| Software, algorithm | Bioconductor package edgeR | *Robinson and Oshlack (2010)* | | |
| Software, algorithm | MACS, version 2.0.1 | *Zhang et al. (2008)* | | |
| Software, algorithm | MUSCLE | *Edgar (2004)* | | |
| Software, algorithm | Integrative Genomics Viewer | *Thorvaldsdóttir et al. (2013)* | | |
| Software, algorithm | MEGA6 | *Tamura et al. (2013)* | | |
| Software, algorithm | Shimadzu Class-VP software (version 6.14 SP1) | Shimadzu, Duisburg, Germany | | |

## Microorganisms, media and cultivation

*A. nidulans* strains were cultivated in *Aspergillus* minimal medium (AMM) at 37°C, 200 rpm (*Brakhage and Van den Brulle, 1995*). When required, supplements were added as follows: arginine (871 µg/ml), *p*-aminobenzoic acid (3 µg/ml) and pyridoxine HCl (5 µg/ml). Pre-cultures were inoculated with $4 \times 10^8$ spores per ml. 10 µg/ml doxycycline was used to induce the *tet*^On-inducible system. *A. nidulans gcnE-3xflag* strain was used for ChIP-seq analysis. For the measurement of orsellinic acid, mycelia of overnight cultures (~16 hr) in AMM were transferred to fresh medium and inoculated with *S. rapamycinicus*, as previously described (*Schroeckh et al., 2009*). RNA extraction for expression analysis during co-cultivation was performed after 3 hr of cultivation; for analysis of the *basR*-overexpression mutant after 6 hr of monoculture, samples for HPLC analysis were taken after 24 hr. *A. sydowii* was cultivated at 28°C, 200 rpm in malt medium (*Scherlach et al., 2010*). For the induction of the *ors* cluster in *A. sydowii*, 48-hr-old precultures were transferred to fresh AMM and inoculated with *S. rapamycinicus* or doxycycline. 10 µg/ml doxycycline was added twice over the course of 48 hr. Samples were taken for LC-MS analysis after 96 hr for *A. sydowii* co-cultivation and after 48 hr for the *A. sydowii basR*-overexpression mutant. For MALDI-MS Imaging analysis, conductive ITO slides (Bruker Daltonics, Bremen, Germany) were coated with 3 ml 0.5% (w/v) AMM agar and incubated at room temperature for 30 min (*Aiyar et al., 2017*; *Araújo et al., 2017*). Identical conditions were ensured by supplementation of all slides with arginine regardless of the fungal genotype. *S. rapamycinicus* was applied by filling 5 ml of a preculture in a tube and by point inoculation of 15 µl of the settled mycelium on the agar. For *A. nidulans*, 500 conidia of wild type and mutants were point inoculated onto the agar. For co-cultivation experiments, both microorganisms were inoculated 1 cm apart from each other. The slides were incubated at 37°C in a Petri dish for 4 days. The slides were dried by incubation in a hybridization oven at 37°C for 48 hr.

## Quantitative RT-PCR (qRT-PCR)

Total RNA was purified with the Universal RNA Purification Kit (Roboklon, Berlin, Germany). Reverse transcription of 5 µg RNA was performed with RevertAid Reverse Transcriptase (Thermo Fisher Scientific, Darmstadt, Germany) for 3 hr at 46°C. qRT-PCR was performed as described before (*Schroeckh et al., 2009*). The *A. nidulans* β-actin gene (*AN6542*) served as an internal standard for calculation of expression levels as previously described (*Schroeckh et al., 2009*). Primers for the amplification of probes are listed in *Supplementary file 4*.

## Preparation of chromosomal DNA and Southern blot analysis

*A. nidulans* genomic DNA was isolated as previously described (*Schroeckh et al., 2009*). Southern blotting was performed using a digoxigenin-11-dUTP-labeled (Jena Bioscience, Jena, Germany) probe (*Schroeckh et al., 2009*).

## ChIP coupled to quantitative RT-PCR (qRT-PCR)

Cultures were grown as described in the cultivation section. After 3 hr, the isolated DNA was cross-linked to proteins as described before (*Boedi et al., 2012*). Powdered mycelium was dissolved in 1 ml of sonication buffer (*Boedi et al., 2012*) and 330 µL aliquots were then subjected to sonication for 30 min with cycles of 2 min maximum intensity followed by a 1 min pause. Sheared chromatin was separated from cell wall debris and incubated with 40 µL of a protein A slurry for 30 min at 4°C on a rotary shaker. A purified 1:10 dilution of the supernatant was then incubated overnight at 4°C with 3 µL of antibody directed against the desired target. Antibodies were precipitated with 40 µL of Dynabeads (Invitrogen, Carlsbad, USA) and were immediately incubated with the sample for 40 min at 4°C on a rotary shaker. Samples were washed three times with low salt buffer followed by washing once with high-salt buffer (*Boedi et al., 2012*). Washed beads were dissolved in 125 µl TES buffer and reverse cross-linked with 2 µL of 0.5 M EDTA, 4 µL of 1 M Tris-HCl pH 6.5 and 2 µL of 1 mg/ml proteinase K for 1 hr at 45°C. Subsequent DNA purification was conducted with a PCR purification kit and samples were eluted in 100 µL of 1:10 diluted elution buffer. The DNA concentration of genes of interest was quantified using qRT-PCR as described above. The antibodies used are the following: mouse monoclonal ANTIFLAG M2 (Sigma-Aldrich, F3165-5MG, Taufkirchen, Germany), rabbit polyclonal anti-histone H3 (Abcam ab1791, Cambridge, UK), rabbit polyclonal histone H3K9ac

(39137, Active Motif, La Hulpe, Belgium)) and rabbit polyclonal anti-acetyl-histone H3 (Lys14) (07–353, Merck Millipore, Darmstadt, Germany).

## Extraction of fungal compounds, HPLC and LC-MS analyses

Culture broth containing fungal mycelium with and without bacteria was homogenized utilizing an ULTRA-TURRAX (IKA-Werke, Staufen, Germany). Homogenized cultures were extracted twice with 100 ml ethyl acetate, dried with sodium sulfate and concentrated under reduced pressure. For HPLC analysis, the dried extracts were dissolved in 1 – 1.5 ml of methanol. Analytical HPLC was performed using a Shimadzu LC-10Avp series HPLC system composed of an autosampler, high pressure pumps, column oven and PDA. HPLC conditions: C18 column (Eurospher 100 – 5 250 × 4.6 mm) and gradient elution (MeCN/0.1% (v/v) TFA ($H_2O$) 0.5/99.5 in 30 min to MeCN/0.1% (v/v) TFA 100/0, MeCN 100% (v/v) for 10 min), flow rate 1 ml min$^{-1}$; injection volume: 50 µL.

The samples of *A. sydowii* were loaded onto an ultrahigh-performance liquid chromatography (LC)–MS system consisting of an UltiMate 3000 binary rapid-separation liquid chromatograph with photodiode array detector (Thermo Fisher Scientific, Dreieich, Germany) and an LTQ XL linear ion trap mass spectrometer (Thermo Fisher Scientific, Dreieich, Germany) equipped with an electrospray ion source. The extracts (injection volume, 10 µL) were analyzed on a 150 x 4.6 mm Accucore reversed-phase (RP)-MS column with a particle size of 2.6 µm (Thermo Fisher Scientific, Dreieich, Germany) at a flow rate of 1 ml/min, with the following gradient over 21 min: initial 0.1% (v/v) HCOOH-MeCN/0.1% (v/v) HCOOH-$H_2O$ 0/100, which was increased to 80/20 in 15 min and then to 100/0 in 2 min, held at 100/0 for 2 min, and reversed to 0/100 in 2 min.

Identification of metabolites was achieved by comparison with an authentic reference. Samples were quantified via integration of the peak area using Shimadzu Class-VP software (version 6.14 SP1).

## MALDI-MS imaging analysis and data processing

Sample preparation and matrix coating were performed as previously described (*Aiyar et al., 2017*). Samples were analyzed (*Aiyar et al., 2017*) in an UltrafleXtreme MALDI TOF/TOF (Bruker Daltonics, Bremen, Germany), in reflector positive mode with the following modifications: 100 – 3000 Da range, 30% laser intensity (laser type 4) and raster width 200 µm. The experiments were repeated three times (2$^{nd}$ and 3$^{rd}$ replicates with 250 µm raster width). Calibration of the acquisition method, spectra procession, visualization, analysis and illustration were performed as described before (*Aiyar et al., 2017*). Chemical images were obtained using Median normalization and weak denoising.

## Resazurin assay

Respiratory activity was measured by reduction of resazurin to the fluorescent dye resorufin. $10^4$ conidia of *A. nidulans* in 100 µL AMM were pipetted into each well of a black 96-well plate. The plate was incubated for 16 hr at 37°C. The pre-grown fungal mycelium was further cultivated in monoculture or with 10 µL of an *S. rapamycinicus* culture. Cultures were further supplemented with 100 µL of AMM containing resazurin in a final concentration of 0.02 mg/ml. Fluorescence was measured (absorption wavelength 560 nm, emission wavelength 590 nm) every 30 min for 24 hr at 37°C in a Tecan fluorometer (Infinite M200 PRO, Männedorf, Switzerland). For all conditions, measurements were carried out in triplicates for each of the two biological replicates. The significance of values was calculated using a two-way ANOVA Test with GraphPad Prism 5 (GraphPad Software Inc., La Jolla, USA).

## ChIP-seq pre-processing

The *A. nidulans* FGSC A4 genome and annotation (version s10-m03-r28) were obtained from the *Aspergillus* Genome Database (AspGD) (*Cerqueira et al., 2014*). The *S. rapamycinicus* NRRL 5491 genome was obtained from NCBI (GI 521353217). Both genomes were concatenated to a fused genome which served as the reference genome for subsequent mapping. Raw ChIP-seq reads were obtained using FastQC v0.11.4. Trimming and filtering were achieved by applying Trim Galore utilizing Illumina universal adapter and phred +33 encoding. Reads were not de-duplicated because the duplication rate was <15% for most libraries. Bowtie2 (version 2.2.4) using default parameters was

employed to map reads to the fused genome. Quantification of reads was carried out using the Bioconductor 'GenomicAlignments' package forming the basis for three subsequent approaches. First, a genome-wide equi-spaced binning across the genome with different resolutions (50 k and 2 k bp bins) counting reads overlapping each bin was applied. Library normalization on bin counts was performed by only considering reads mapping to the *A. nidulans* genome. Second, reads overlapping genes were counted, using the AspGD (*Cerqueira et al., 2014*) annotation. They formed the basis for the subsequent DCS analysis (see below). Third, average profile plots to assess relative histone distributions around TSS and TTS were generated using the bioconductor package regioneR (*Gel et al., 2016*).

## DCS analysis

To identify genes exhibiting differences in their chromatin state, we employed the bioconductor package edgeR (*Robinson and Oshlack, 2010*) originally developed for RNA-seq differential expression analysis. The ChIP-seq data follow the same pattern, that is, negative binomial distribution of reads. Library normalization was achieved with the trimmed mean of M values method (*Robinson and Oshlack, 2010*) based only on *A. nidulans* gene counts for calculating the effective library sizes, not taking into account reads mapping to *S. rapamycinicus* which would otherwise artificially influence the effective library size. Comparisons were made between libraries for all ChIP targets separately obtained from monocultures of *A. nidulans* and co-cultures with *S. rapamycinicus*. These targets were H3, H3K9ac and H3K14ac. Results including normalized read counts (RPKM), statistics and LFCs are reported in *Supplementary file 1*. Normalized counts and LFCs were also further used for comparisons with the corresponding microarray-based gene expression and the calculated LFCs.

## MACS analysis

Candidate peaks were identified using two methods: a differential binding analysis (EdgeR) and a peak-calling approach (MACS, version 2.0.1) (*Zhang et al., 2008*). The peak caller performed several pairwise comparisons between samples with the same antibody and between different conditions in order to retrieve the peaks with significant change of ChIP signal indicating differential binding for that particular comparison. The program kept the track of different replicates, the signal was reported per million reads and produced a BED format track of the enriched regions, other parameters were used with default values. The BED files were subsequently converted to Big Wig format for visualization through the tool Integrative Genomics Viewer (*Thorvaldsdóttir et al., 2013*).

## Generation of *A. nidulans* deletion strains

The transformation cassettes for the *basR* and *AN8377* deletion strains were constructed as previously described (*Szewczyk et al., 2007*). Approximately ~1000 bp sequences homologous to the regions upstream and downstream of *basR* and *AN8377* were amplified and fused to the *argB* deletion cassette (*Schroeckh et al., 2009*). Transformation of *A. nidulans* was carried out as described before (*Ballance and Turner, 1985*).

## Generation of inducible *A. nidulans* and *A. sydowii basR*-overexpressing strains

For overexpression of *basR*, the tetracycline-controlled transcriptional activation system ($tet^{On}$) was used (*Helmschrott et al., 2013*). The *basR* gene sequences together with their ~ 1000 bp flanking regions were amplified from *A. nidulans* and *A. sydowii* genomic DNA. The $tet^{On}$-system was amplified from plasmid pSK562. All DNA fragments were assembled using NEBuilder HiFi DNA Assembly Master Mix (New England Biolabs, Frankfurt, Germany). The *A. nidulans pabaA*1 gene was used as a selectable marker to complement the *p*-aminobenzoic acid auxotrophy of the *A. nidulans* Δ*basR* mutant. For *A. sydowii*, the *Aspergillus oryzae hph* cassette was used as the selectable marker. 200 µg/ml hygromycin (Invivogen, Toulouse, France) was used for selection of transformant strains.

## Phylogenetic analysis

The amino-acid sequences for the two Myb-like transcription factors from *A. nidulans* (AN7174 (*basR*) and *AN8377*) and Bas1 from *S. cerevisiae* were used for a Blast search in the UniProtKB

database. For each sequence, the first 50 hits were retrieved. All hits were grouped together, and redundant and partial sequences removed. The obtained 54 hits were first aligned using MUSCLE (*Edgar, 2004*). The phylogenetic tree was obtained using the Maximum Likelihood method contained in the MEGA6 software facilities (*Tamura et al., 2013*).

## Measurement of amino acids

Amino acids were extracted from 10 mg samples with 1 ml of methanol and the resulting extract was diluted in a ratio of 1:10 (v:v) in water containing the $^{13}$C, $^{15}$N labeled amino-acid mix (Isotec, Miamisburg, Ohio, USA). Amino acids in the diluted extracts were directly analyzed by LC-MS/MS as described, but with the modification that an API5000 mass spectrometer (Applied Biosystems, Foster City, California, USA) was used (*Docimo et al., 2012*).

## cDNA library construction and sequencing

Total RNA was isolated as described for qRT-PCR analysis, from three replicates of *A. nidulans* cultivated with and without *S. rapamycinicus* and from the inducible *A. nidulans basR*-overexpression strain with and without the addition of doxycycline. Samples were taken after six hours of co-cultivation or addition of doxycycline to induce the *basR* overexpression. Total RNA from the three replicates was pooled and 9 µg of RNA were processed for the library preparation. Library construction, Illumina next-generation sequencing, and the mapping and normalizing of the transcript reads were performed by StarSEQ GmbH (Mainz, Germany). Transcript levels were normalized by counting the number of transcripts per million (TPM) (*Wagner et al., 2012*).

## Availability of data and materials

ChIP-seq data were deposited in the ArrayExpress database at EMBL-EBI (www.ebi.ac.uk/arrayexpress) under accession number E-MTAB-5819. The code for data processing and analysis can be obtained from https://github.com/seb-mueller/ChIP-Seq_Anidulans (*Müller, 2018*; copy archived at https://github.com/elifesciences-publications/ChIP-Seq_Anidulans).

## Acknowledgements

We thank Christina Täumer and Karin Burmeister for excellent technical assistance and Sven Krappmann (Friedrich-Alexander University, Erlangen-Nürnberg, Germany) for kindly providing plasmid pSK562. Financial support by the Deutsche Forschungsgemeinschaft (DFG)-funded excellence graduate school Jena School for Microbial Communication (JSMC), the International Leibniz Research School for Microbial and Biomolecular Interactions (ILRS) as part of the JSMC, the DFG-funded Collaborative Research Center 1127 ChemBioSys (projects B01, B02, INF and B07), the BMBF-funded project DrugBioTune in the frame of Infectcontrol2020, the Leibniz Research Cluster in the frame of the BMBF Strategic Process Biotechnology 2020+ and the European Research Council for a Marie Skłodowska-Curie Individual Fellowship (IF-EF; Project reference 700036) to María García-Altares is gratefully acknowledged. Chromatin work was funded by grants SFB-F3703 of the Austrian Science Fund (FWF) and BiMM-K3-G-2/026-2013 of the Lower Austria Science Fund (NFB) to Joseph Strauss.

## Additional information

### Competing interests

Axel A Brakhage: Reviewing editor, *eLife*. The other authors declare that no competing interests exist.

### Funding

| Funder | Grant reference number | Author |
| --- | --- | --- |
| Leibniz-Institut für Naturstoff-Forschung und Infektionsbiologie – Hans-Knöll-Institut | International Leibniz Research School for Microbial and Biomolecular Interactions (ILRS) | Juliane Fischer Sebastian Y Müller Axel A Brakhage |

| Deutsche Forschungsge-meinschaft | SFB 1127 - ChemBioSys | Tina Netzker<br>Nils Jäger<br>Jonathan Gershenzon<br>Ekaterina Shelest<br>Thorsten Heinzel<br>Christian Hertweck<br>Axel A Brakhage |
|---|---|---|
| Bundesministerium für Bildung und Forschung | DrugBioTune - Infectcontrol2020 | Maria C Stroe<br>Axel A Brakhage |
| Horizon 2020 Framework Programme | IF-EF; Project reference 700036 | María García-Altares |
| Deutsche Forschungsge-meinschaft | Jena School for Microbial Communication (JSMC) | Francesco Pezzini<br>Mario KC Krespach<br>Axel A Brakhage |
| Bundesministerium für Bildung und Forschung | Leibniz Research Cluster - BMBF Strategic Process Biotechnology 2020+ | Vito Valiante |
| Austrian Science Fund | SFB-F3703 | Joseph Strauss |
| Lower Austria Science Fund (NFB) | BiMM-K3-G-2/026-2013 | Joseph Strauss |

The funders had no role in study design, data collection and interpretation, or the decision to submit the work for publication.

## Author contributions

Juliane Fischer, Conceptualization, Investigation, Visualization, Methodology, Writing—original draft, Writing—review and editing; Sebastian Y Müller, Data curation, Formal analysis, Investigation, Visualization, Methodology, Writing—original draft, Writing—review and editing; Tina Netzker, Investigation, Visualization, Writing—original draft; Nils Jäger, Investigation, Visualization, Writing—review and editing; Agnieszka Gacek-Matthews, Resources, Investigation, Writing—review and editing; Kirstin Scherlach, Michael Reichelt, Mario KC Krespach, Investigation; Maria C Stroe, María García-Altares, Hanno Schoeler, Investigation, Visualization; Francesco Pezzini, Formal analysis; Jonathan Gershenzon, Resources, Investigation; Ekaterina Shelest, Formal analysis, Writing—review and editing; Volker Schroeckh, Formal analysis, Supervision, Writing—review and editing; Vito Valiante, Formal analysis, Visualization, Writing—review and editing; Thorsten Heinzel, Supervision, Writing—review and editing; Christian Hertweck, Investigation, Writing—review and editing; Joseph Strauss, Conceptualization, Resources, Writing—review and editing; Axel A Brakhage, Conceptualization, Data curation, Supervision, Funding acquisition, Writing—original draft, Project administration, Writing—review and editing

## Author ORCIDs

Sebastian Y Müller (iD) http://orcid.org/0000-0002-0148-7020
Tina Netzker (iD) http://orcid.org/0000-0003-4845-9366
Francesco Pezzini (iD) http://orcid.org/0000-0002-2892-7997
Michael Reichelt (iD) http://orcid.org/0000-0002-6691-6500
Volker Schroeckh (iD) http://orcid.org/0000-0001-5716-2957
Vito Valiante (iD) http://orcid.org/0000-0002-4405-169X
Joseph Strauss (iD) http://orcid.org/0000-0003-0474-6267
Axel A Brakhage (iD) http://orcid.org/0000-0002-8814-4193

## Decision letter and Author response

Decision letter https://doi.org/10.7554/eLife.40969.034
Author response https://doi.org/10.7554/eLife.40969.035

## Additional files

### Supplementary files

• Supplementary file 1. Summary of DCS analysis of H3K9ac between *A. nidulans* monoculture and co-cultivation with *S. rapamycinicus*. Genes that are aceylated at significantly higher levels are marked in red whereas those that are acetylated at lower levels are marked in blue.
DOI: https://doi.org/10.7554/eLife.40969.017

• Supplementary file 2. List of selected genes with differentially acetylated H3K9 and different expression.
DOI: https://doi.org/10.7554/eLife.40969.018

• Supplementary file 3. Summary of RNA-seq data. List of selected differentially expressed genes in *A. nidulans* wild type and in the *A. nidulans* tet^On-*basR* mutant in response to *S. rapamycinicus* or after doxycycline addition.
DOI: https://doi.org/10.7554/eLife.40969.019

• Supplementary file 4. List of primers used in this study.
DOI: https://doi.org/10.7554/eLife.40969.020

• Supplementary file 5. Summary of ChIP-seq data.
DOI: https://doi.org/10.7554/eLife.40969.021

• Transparent reporting form
DOI: https://doi.org/10.7554/eLife.40969.022

### Data availability

ChIP-seq data were deposited in the ArrayExpress database at EMBL-EBI (www.ebi.ac.uk/arrayexpress) under accession number E-MTAB-5819.

The following dataset was generated:

| Author(s) | Year | Dataset title | Dataset URL | Database and Identifier |
|---|---|---|---|---|
| Sebastian Y Müller, Juliane Fischer | 2018 | ChIP-seq data | https://www.ebi.ac.uk/arrayexpress/experiments/E-MTAB-5819/ | EMBL-EBI ArrayExpress, E-MTAB-5819 |

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

## Appendix 1

DOI: https://doi.org/10.7554/eLife.40969.023

## Supplementary results

### Details of the ChIP-seq analysis

After first examination, we found that a significant proportion of co-incubated library reads originated from *S. rapamycinicus*. A fused genome concatenating the *A. nidulans* and the *S. rapamycinicus* genomes was generated. About 90-98% of the reads mapped against the fused genome (**Supplementary file 5**), which suggested a high quality of sequencing data. This assumption was confirmed by examining the quality of libraries using FastQC (data not shown). As indicated, the coverage was substantially higher on the *S. rapamycinicus* genome with only ~20 % mapping to the *A. nidulans* genome (see **Supplementary file 5**). Expectedly, this ratio has shifted considerably towards the *A. nidulans* genome for histone targeting ChIP libraries (**Figure 1a**) going up from 20 % to around 90 % for all H3, H3K9 and H314 libraries validating correct antibody enrichment as *S. rapamycinicus* is devoid of histones. However around 10 % of reads were still mapping to the *S. rapamycinicus* genome which might be due to imperfect antibody specificity. Coverage depth deviations were accounted for by only considering reads originating from *A. nidulans*. This allowed for calculation of library size factors used for library normalization. To assess antibody specificity, we calculated the fraction of reads mapping to mitochondria, which do not contain histones. The control library amounted to about 0.25 % of reads as opposed to about 0.01– 0.03 % for H3, H3K9ac, H3K14ac libraries constituting a 10-fold enrichment. As a background control we used ChIP material obtained from anti-FLAG-tag antibody precipitates of a non-tagged fungal wild-type strain co-cultivated under the same conditions with *S. rapamycinicus*.

We quantified the relative library proportions which amounted to 8–17 % of H3, H3K14ac and H3K9ac as well as up to ~65 % for background reads of co-incubated libraries mapped to *S. rapamycinicus* (see **Supplementary file 5**). However, the background read proportions might not necessarily reflect actual gDNA ratios of both species in the co-cultivation due to various potential biases. To examine read distribution for each library, we counted mapped reads within equally spaced bins along the fused genome for different resolutions (see methods and **Figure 1a and b**). As expected, background reads (upper panel of **Figure 1a**) were evenly distributed across the genome reflecting nonspecific targeting of particular areas. The fused genome further enabled for controlling correct co-incubation conditions since no reads should be mapping to *S. rapamycinicus* in non-co-incubated samples as can be seen in **Figure 1a** in the right panels (blue lines). The co-incubated samples exhibit an increased coverage in the middle of the *S. rapamycinicus* genome which might be due to sequence biases or enriched DNA caused by the replication origin (*oriC*) located in this region (**Jakimowicz et al., 1998**). Further, there were coverage dips in the middle of the fungal chromosomes (see **Figure 1a**), which were most likely due to incomplete assembly around the centromeres. They are characterized by long 'N' stretches (**Ekblom and Wolf, 2014**).

### Changes of H3K9 and H3K14 acetylation profiles in *A. nidulans* in response to *S. rapamycinicus*

As reported in the manuscript, the genome-wide H3K9 and H3K14 chromatin landscape of *A. nidulans* were determined. There was also a drop-off in all libraries at the chromosome arms, which was most likely caused by the bordering bins being shorter and therefore they account for less reads. Since gene density also varies across the genome (lower panel of **Figure 1a**), we addressed the question whether this correlates with the intensity of the investigated chromatin states. To this end, we calculated the spearman correlation to correlate read counts and gene counts among the 50k bins. As expected, there was almost no correlation between the background and the genes (r=0.09). However, for H3 we found it to be rather high

(r=0.37) (*Appendix 1—figure 4*). Since the used bin size is large, this could point at global H3 occupancy to be higher for high gene density regions such as euchromatin and low H3 occupancy for heterochromatin. Noteworthy, the correlation between read and gene density was found to be lower for H3K14ac and H3K9ac (r=0.14 and 0.15 respectively), which might indicate a more subtle regulatory mechanism for those marks targeting individual genes as opposed to larger domains. Notably, the highest correlation was found between the two acetylation marks (r=0.53) hinting at some potential cross-talk or common regulation between them (*Appendix 1—figure 4*).

## Chromatin profiles at translation start sites and translation termination sites

We assessed the location of H3K9ac, H3K14ac and histone H3 relative to promoters and gene bodies by plotting the average read count frequency for all genes to either the TSSs or TTSs (*Appendix 1—figure 1*) (*Yu et al., 2015*). Due to missing information about the 5' and 3' transcriptional start and stop sites, we used translational start and stop sites for transcription start and stop sites, respectively, as surrogates. The results obtained apply to both *A. nidulans* in mono- and co-culture with the bacterium. According to *Kaplan et al. (2010)*, the peaks correspond to highly positioned nucleosomes relative to the TSSs with a nucleosome-free region (NFR) directly upstream of the TSS. H3K9ac and H3K14ac showed highest enrichment for the first and second nucleosomes up- and downstream of the TSS and drastic reduction downstream of the third nucleosome after the TSS (*Appendix 1—figure 1*). Reduced occupancy of unmodified histone H3 was observed at the TSS (*Kaplan et al., 2010*). Towards the 3' end of genes, histone H3 occupancy gradually increased, which was accompanied by a decrease in acetylation. Plotting of differentially acetylated H3K9ac against H3K14ac showed a strong correlation between the localization of the two modifications (*Figure 2—figure supplement 2a*). Acetylation is generally described as a transcription activating mark (*Gacek and Strauss, 2012*). To test for this general assumption we correlated our acetylation data to microarray data generated under the same condition (*Nützmann et al., 2011*) (*Figure 2—figure supplement 2b*). This allowed us to compare the log-fold changes (LFCs) of the differential chromatin states with the LFCs calculated from the RNA expression data. H3K9ac correlates with the differentially expressed genes with a coefficient of 0.5 in contrast to H3K14ac (-0.05) and histone H3 (-0.01) for which no detectable dependency was observed (*Appendix 1—figure 3*). Similarly, *Appendix 1—figure 3* shows the same trend, i.e., a correlation of 0.2 for gene expression changes *versus* H3K9ac changes. This finding was expected since the calculation included all genes most of which did not change. To determine the correlation of acetylation at the TSS and the TTS according to the grade of expression of the genes, we separated the differentially expressed genes into four quartiles (q1 lower 25 %, q2 the medium lower 25–50 %, q3 are the medium higher 50-75 %, q4 higher 25 %). The increase of acetylation at the TSS correlated with the expression level of genes (*Appendix 1—figure 6a and c*). Decreased expression coincided with an increase of histone H3 at the TSS (*Appendix 1—figure 6e*).

## Reduced intracellular amino acid concentration in response to the bacterium

The increased expression of *cpcA* and other genes involved in amino acid biosynthesis implied a reduced availability of amino acids in the cell upon bacterial-fungal co-cultivation. Therefore, we measured the internal amino acid pool in *A. nidulans* both grown in monoculture and with *S. rapamycinicus* (*Appendix 1—figure 7*). As an additional control and to further confirm the specificity of the interaction, the fungus was co-cultivated with *Streptomyces lividans*, which does not induce the *ors* gene cluster. As shown in *Appendix 1—figure 7*, significantly reduced levels of glutamine, histidine, phenylalanine, asparagine, threonine and reduced metabolism of arginine, which was supplemented to the medium, were observed. The monoculture of *A. nidulans*, co-cultivation of *A. nidulans* with *S. lividans* as well as the addition

of *S. rapamycinicus* after 24 hours of fungal cultivation served as controls. All of the controls showed comparable amino acid levels.

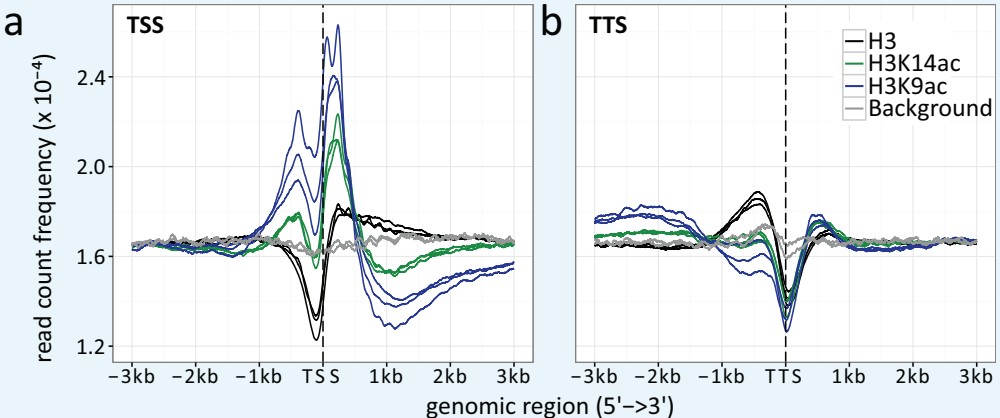

**Appendix 1—figure 1.** Read count frequencies for (**a**) TSSs and (**b**) TTSs. Lines correspond to the relative enrichment of ChIP signal strength relative to the TSS/TTS averaged across all genes. ChIP-seq read count serves as a surrogate for signal strength (see methods for further details). Compared were the enrichment of histone H3 (black), H3K9ac (blue), H3K14ac (green) and the background control (gray) over an average of all TSSs and TTSs. The enrichment curves for all biological replicates are given, indicated by multiple lines per enrichment target.
DOI: https://doi.org/10.7554/eLife.40969.024

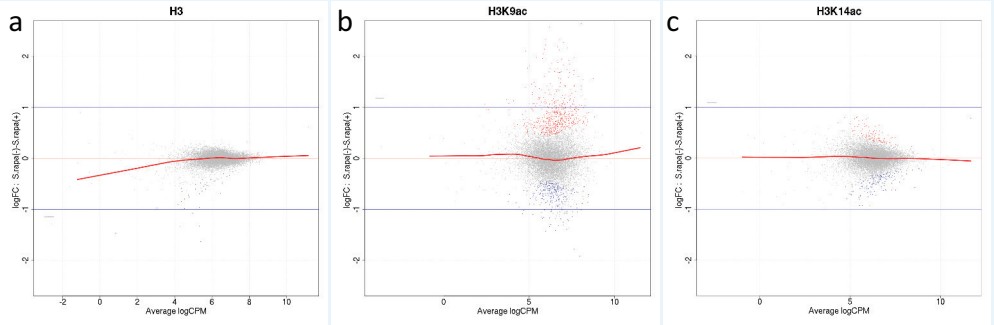

**Appendix 1—figure 2.** Mean average (MA) plots comparing normalized ChIP-seq LFCs H3 (Cterm) (**a**), H3K14ac (**b**) and H3K9ac (**c**) of each gene between *A. nidulans* monoculture and co-culture for antibodies used in this study. The y-axis indicates LFCs of ChIP signal between mono- and co-culture for each gene which corresponds to the dots. The x-axis indicates mean intensity of ChIP signal of both conditions. Genes were colored according to DCS outcome with gray indicating genes with no significant change of ChIP signal, red and blue indicate genes showing respective higher or lower signal in co-culture.
DOI: https://doi.org/10.7554/eLife.40969.025

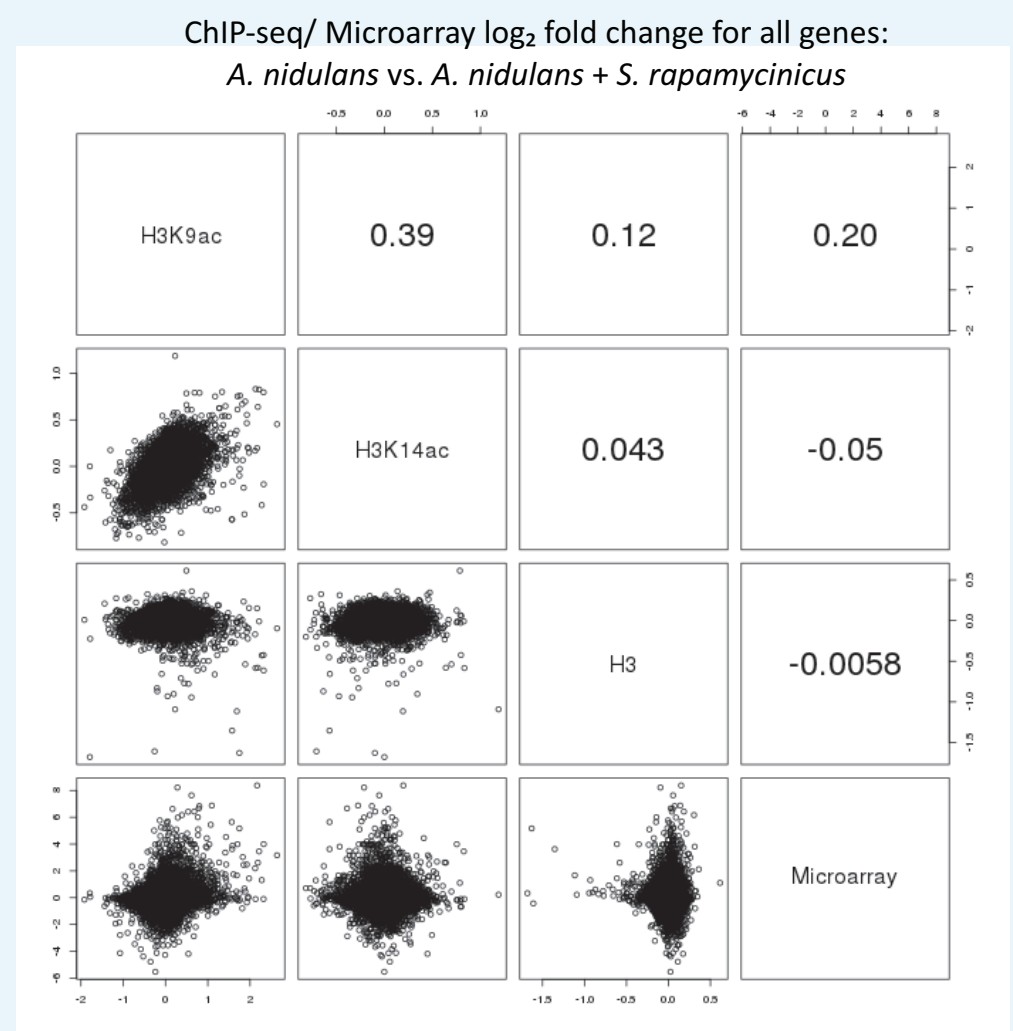

**Appendix 1—figure 3.** Correlation of data points for LFCs of ChIP-seq with LFCs of microarray data for all *A. nidulans* genes, depicting single data points and the correlation coefficient.

DOI: https://doi.org/10.7554/eLife.40969.026

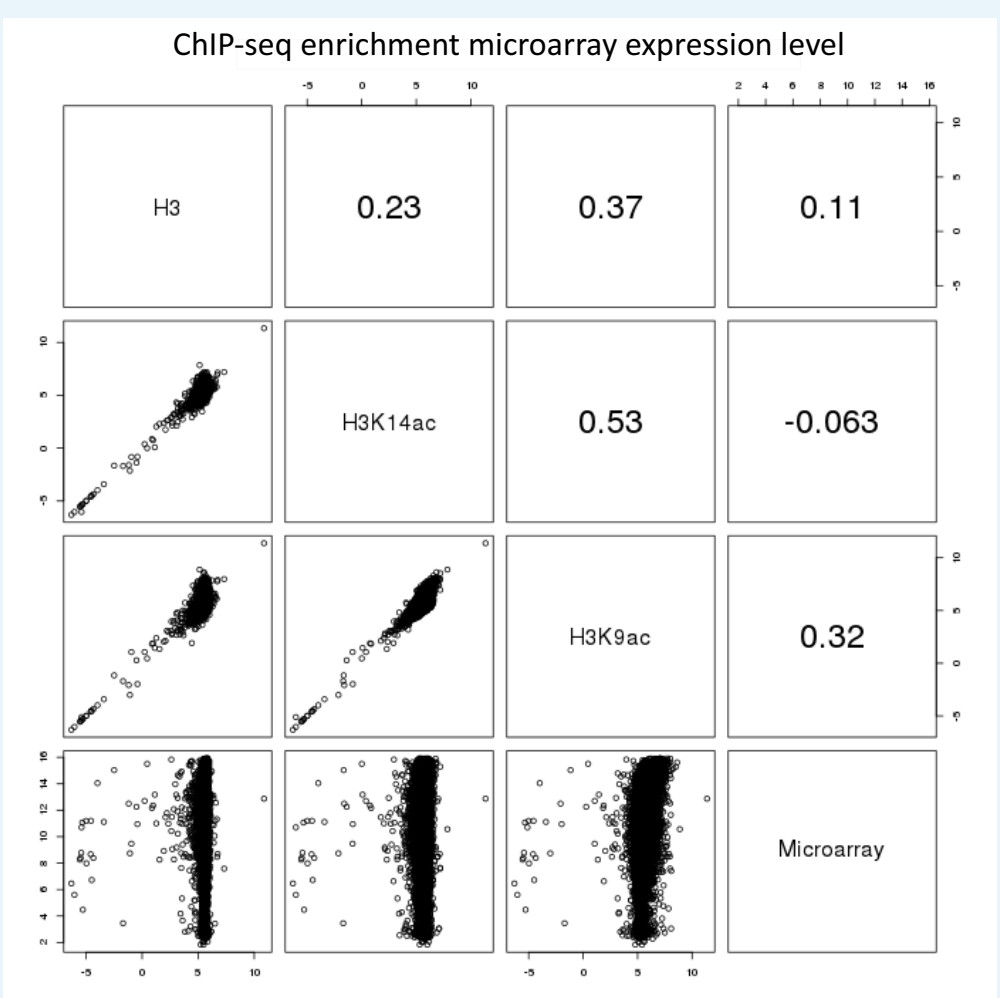

**Appendix 1—figure 4.** Pairwise comparison of ChIP-seq and microarray intensities of all genes in *A. nidulans* monoculture. The numbers resemble the correlation coefficient for the respective comparison. Intensity defines enrichment of number of reads per gene.

DOI: https://doi.org/10.7554/eLife.40969.027

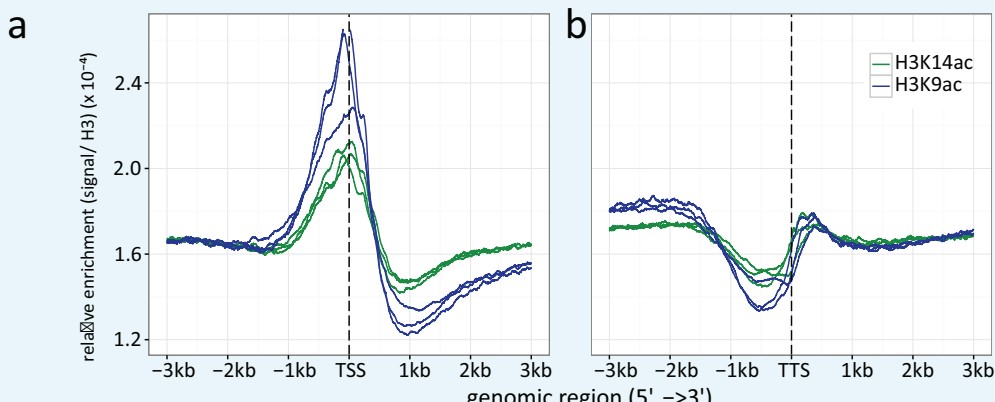

**Appendix 1—figure 5.** Histone H3 normalized read count frequencies for H3K9ac (green) and K14 ac (blue) at the (**a**) TSSs and (**b**) TTSs. The enrichment is given in signal to H3 ratio. Multiple lines per ChIP target resemble the three independent biological replicates.

DOI: https://doi.org/10.7554/eLife.40969.028

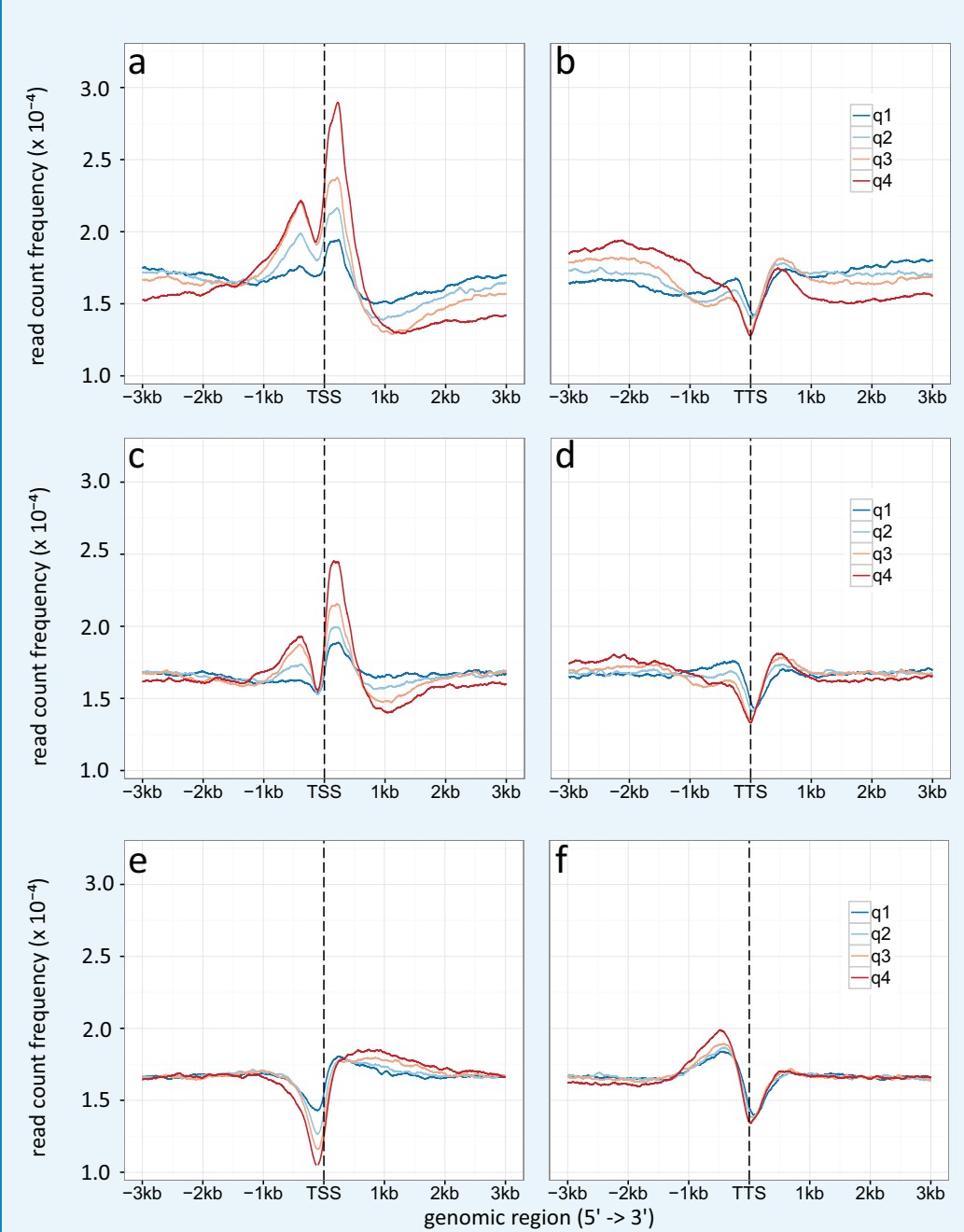

**Appendix 1—figure 6.** Density plot of TSSs (**a, c, e**) and TTSs (**b, d, f**) given for different gene expression levels (q1-q4). (**a, b**) Specific enrichment of H3K9ac, (**c, d**) H3K14ac and (**e, f**) H3 is given in read count frequency. q1 are the lower 25 %, q2 the medium lower 25 – 50 %, q3 are the medium high 50-75 %, q4 the higher 25 %.

DOI: https://doi.org/10.7554/eLife.40969.029

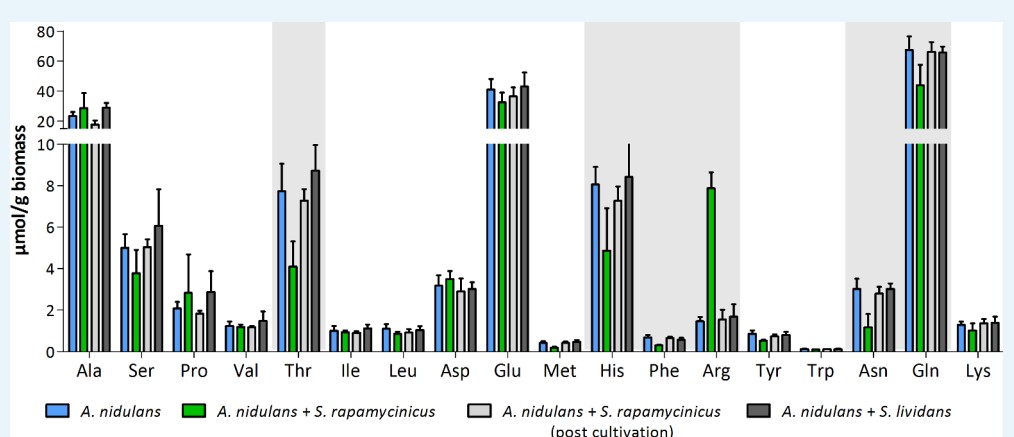

**Appendix 1—figure 7.** Intracellular amino acid concentration of *A. nidulans* in monoculture and co-culture with *S. rapamycinicus*. Co-cultivation with *S. lividans* and addition of *S. rapamycinicus* after 24 hours of cultivation served as negative controls. Furthermore, before extraction of amino acids the fungus (post cultivation) was also supplemented with *S. rapamycinicus* to exclude a bias resulting from bacterial amino acids. Threonine, histidine, phenylalanine, arginine, asparagine and glutamine showing different concentrations in co-culture compared to monoculture are highlighted in gray.

DOI: https://doi.org/10.7554/eLife.40969.030

