## [Decision Letter]

[Editors’ note: a previous version of this study was rejected after peer review, but the authors submitted for reconsideration. The first decision letter after peer review is shown below.]

Thank you for submitting your work entitled "Bacterium-triggered remodeling of chromatin identifies BasR, a novel regulator of fungal natural product biosynthesis" for consideration by *eLife*. Your article has been reviewed by three peer reviewers, one of whom is a member of our Board of Reviewing Editors, and the evaluation has been overseen by a Senior Editor. The reviewers have opted to remain anonymous.

Our decision has been reached after consultation between the reviewers. Based on these discussions and the individual reviews below, we regret to inform you that your work in its current form will not be considered further for publication in *eLife*. Nevertheless, we appreciate the topic and find the work in principle very interesting. Thus, if you choose not to send the work as is elsewhere, but rather revise the study, *eLife* would be prepared to review the work again. It would, however, be treated as a new submission, although we would try to retain the same reviewers.

Summary:

This manuscript presents three connected "stories". The first story is the generation of genome-wide H3K9 and H3K14 acetylation maps in *A. nidulans* in the absence and presence of the actinobacterium, *S. rapamycinicus*. The second story is on the regulation of specific biosynthetic gene clusters (BGCs), where the authors focus again on the orsellinic BGC, which was the subject matter of three previous publications by the same group. The third story includes the truly novel part of the manuscript but unfortunately gets the least amount of space. This is the interesting finding that the (supposedly intimate) contact, shown in a previous paper, of the actinobacterium with *Aspergillus hyphae* somehow decrease the available nitrogen, causing an imbalance in amino acid uptake or utilization which triggers the "General Control" or "Cross-Pathway Control" system, regulated by Gcn4/CpcA- and Bas-type TFs. The authors argue that one of the putative Bas1-related TFs, now named BasR, is responsible for the bacterium-induced control.

Essential revisions:

In general, all reviewers were concerned about the novelty of this study as written and deliberated extensively on whether to "reject" or "revise". In the first story, the novelty of the study lies in the genome-wide chromatin maps generated in the presence of the bacterium, whereas previously only small regions were studied. However, the general features of chromatin regulation by themselves (i.e., upregulation by increased H3K9ac and some increase in H3K14ac) are nothing new as similar studies have been done in other fungi (yeast, *Fusarium*), plants and animals, both in the presence or absence of stressors. In the second story, the connection between *S. rapamycinicus*, acetylation, and orsellinic acid production has previously been reported, and the authors do not report incisive analyses on novel clusters. Instead they focus, quite unaccountably, on the one cluster that is down- instead of up-regulated (*eas*); so no new products are identified or characterized here. The third story, which has the potential to be the most novel, unfortunately gets the least amount of space and what the authors report is a phenomenological description of the pathway uncovered with no mechanistic studies other than genetic deletion analysis.

Thus, the reviewers' general recommendation is that the authors should focus their manuscript and substantially expand either the second story or the third story (in conjunction with the first). If the authors decide to expand the second story, they should describe the effects on novel BGCs and compounds. If the authors decide to expand on the third story, they should further describe the BasR pathway and how it may be activated.

1) The authors show that one of the putative Bas1-related TFs, now named BasR, is responsible for the bacterium-induced control but they completely disregard potential function of the second TF, *AN8377*, based on the apparently unchanged level of H3K9ac in the gene (which is not specifically shown). While *AN7174* (BasR) has very restricted distribution in fungi (although there seem to be homologs in *Saprolegnia, Malassezia* and *Ustilago*; also according to FungiDB), *AN8377* is the more wide-spread factor. The authors do not mention anything about this Myb/SANT domain protein and its function at all. What is shown is a phenomenological description of the pathway uncovered with no mechanistic studies other than genetic deletion analysis. For example, instead of using a rapamycin-deficient mutant of a *Streptomyces* species that presumably did not cause induction of Gcn5, why wasn't rapamycin simply added to the medium to test whether it was required to elicit the outcome (especially since yeast *gcn5* mutants are sensitive to rapamycin). Perhaps this had been done before but then the logic of the experiment with the mutant of an unrelated species is unclear. There is also not direct evidence for BasR regulating either the genes in amino acid metabolism affected by cross-pathway control or the final target genes in clusters, e.g. by binding to motifs in their promoters.

2) If the authors focus their manuscript on the identification of circuits that regulate amino acid availability in *Aspergillus*, they should give their manuscript a new title. "Chromatin remodeling" specifically refers to the movement of nucleosomes, either sliding, removal or replacement, by a group of large ATPases. The authors are discussing "chromatin modifications" by a lysine acetyltransferase. This is not the same thing as "remodeling". These changes in modifications did not per se result in the identification of BasR, and that BasR directly "regulates" natural product biosynthesis also has not been shown, though it may very well affect this indirectly. Also, BasR is not novel, but distantly related to Bas1/2 in *S. cerevisiae*.

3) The reviewers appreciate the authors wanting to get good ChIP data on this interaction which really is where the field needs to move. However, the pulldown with antibodies against histone should not pick up any bacterial DNA. Unfortunately, it brought down lots of bacterial DNA. One possibility is that their bacteria have something like Protein A, which would indiscriminately bind the IgG heavy chain. The "fused genome" designation is basically accounting for this contamination which has to have some impact on the results. The reviewers wonder if the authors could have mixed the fungus with cell fragments of the bacterium or some other method to reduce this contamination. The reviewers like the idea of DCA, but are not sure how one can use the data, because of the contamination from the bacterial DNA. We presume that the fungal information is still viable, but would worry that you're introducing biases in library prep, etc., especially when comparing to the fungal samples where 100% of the reads map to *Aspergillus*. More controls might help. Would this ChIP approach bring down bacterial contamination if other bacteria were used? Maybe some focus on methods would be good to work out a cleaner result.

[Editors’ note: what now follows is the decision letter after the authors submitted for further consideration.]

Thank you for resubmitting your work entitled "Chromatin mapping identifies BasR, the regulatory node of bacteria-triggered production of fungal secondary metabolites" for further consideration at *eLife*. Your revised article has been favorably evaluated by Detlef Weigel as the Senior Editor, and three reviewers, one of whom is a member of our Board of Reviewing Editors.

The manuscript has been improved but there are some remaining issues that need to be addressed before acceptance, as outlined below:

1) Title: change "the regulatory node" to "a key regulatory node" since there are likely other regulatory nodes that contribute to *S. rapaminicus*-induced SM cluster activation (your data supports this hypothesis: 3/8 differentially acetylated SM gene clusters were not differentially transcribed in response to *basR* overexpression).

2) The phylogenies of *basR* depicted on Figure 6 and Figure 6—figure supplement 1 look different. Please explain why.

3) “The changes of H3K14ac in Figure 1 are therefore likely due to nucleosome rearrangements towards the translation start sites (TSS) rather than increased amounts of this modification”: Why does this hypothesis only apply to H3K14ac but not H3K9ac as well?

4) The authors start the Abstract and Introduction in such a way that suggests that *S. rapamycinicus* is directly targeting the epigenetic machinery in *A. nidulans*. Particularly when you say "In line…" in the Abstract, as well as give examples of the secretion of methyltransferases. Is this how you believe the change in chromatin is occurring? There is not enough data presented in this work to support this.

5) Abstract, last sentence: I believe you are missing the word "the" between "as…regulatory node".

6)Subsection “Genome-wide profiles of H3K9 and H3K14 acetylation in *A. nidulans* change upon co-cultivation with *S. rapamycinicus*”, last paragraph: Could you define what you mean by "chromatin domain" (approx. size)?

7) Subsection “The transcription factor BasR is the central regulatory node of bacteria-triggered SM gene cluster regulation”, fourth paragraph: You performed an RNAseq experiment with the overexpression of *basR*, and describe the changes in secondary metabolite gene cluster expression. Do you see any changes in the other phenotypes observed in the co-culture with *S. rapamycinicus*? It would be nice to know if there is a decrease in genes associated with nitrogen metabolism and mitochondrial function (Figure 3). This would expand the role of *basR*. Also, does deletion or overexpression of *basR* influence *gcnE* or other members of the Saga/Ada complex expression?

8) Figure 5: Here you illustrate the levels of expression of the *ors* cluster and *basR*, as well as the relative levels of the *ors* cluster products. You mention the "leakiness" of the *tet*^On^ promoter, and we can see an increase in expression of *basR* grown without doxycycline. This level of expression looks similar to that in the co-culture, and yet do not see an increase above what is typically seen of the *ors* cluster. Do you have a hypothesis as to why this is?

9) Subsection “The presence of BasR in fungal species allows forecasting the inducibility of *ors*-like gene clusters by *S. rapamycinicus*”: Earlier in the manuscript you mention the "leakiness" of the *tet*^On^ promoter when overexpressing *basR* in *A. nidulans*. Is this the case in *A. sydowii*? If not, it would be really nice to see the loss of *ors* products in the co-culture of *S. rapacycinicus* and *A. sydowii tet*^On^-*basR* without doxycycline, demonstrating the conserved role of *basR* in the interaction of the two microbes.

10) How was the anti-H3 antibody validated? This particular antibody is great for western blots but does not always work for ChIP (see ENCODE histone antibody database, http://compbio.med.harvard.edu/antibodies/targets/12 and the histone validation service by the Strahl lab and company: http://www.histoneantibodies.com/). In our hands this H3 antibody does not work for ChIP. The same concern applies to the anti-H3K9ac and anti-H3K914ac antibodies; Active Motif 39161 does not show up as a validated ChIP antibody in either database, and no catalog number is given for the second antibody.

11) The model figure is interesting but some of the interactions shown have not been established. The authors show arrows implying that BasR directly regulates the clusters but that has not been shown by experiment. Are binding sites found in the promoters of cluster genes?

12) There is still relatively little mechanistic information about the regulatory cascade from stressor to induction of cluster transcription. How the switch from normal to imbalanced or depleted nitrogen occurs in presence of *Streptomyces* is the big question, and how GcnE and BasR are linked in *Aspergillus* is still not clear.

13) Subsection “Increased gene expression directly correlates with histone H3K9 acetylation”: The low correlation between active gene transcription and acetylation at H3K14 confirmed earlier results – a citation is needed here.

14) "did not affect the induction of the *ors* gene cluster, but on the other hand the artificial inducer of the CPC system 3-AT DOES (Sachs, 1996), it…" – this sentence needs a "does" or something similar to make it clear.

---

## [Author Response]

[Editors’ note: the author responses to the first round of peer review follow.]

We have followed the suggestions of the reviewers and have included additional experiments and redrafted the manuscript. According to your general recommendation, we have expanded the story about the Myb-like transcription factor BasR (e.g. dependency on GcnE, regulation of other secondary metabolite gene cluster) and concurrently we condensed the part about the chromatin profiles.

Essential revisions:In general, all reviewers were concerned about the novelty of this study as written and deliberated extensively on whether to "reject" or "revise". In the first story, the novelty of the study lies in the genome-wide chromatin maps generated in the presence of the bacterium, whereas previously only small regions were studied. However, the general features of chromatin regulation by themselves (i.e., upregulation by increased H3K9ac and some increase in H3K14ac) are nothing new as similar studies have been done in other fungi (yeast, Fusarium), plants and animals, both in the presence or absence of stressors. In the second story, the connection between S. rapamycinicus, acetylation, and orsellinic acid production has previously been reported, and the authors do not report incisive analyses on novel clusters. Instead they focus, quite unaccountably, on the one cluster that is down- instead of up-regulated (eas); so no new products are identified or characterized here. The third story, which has the potential to be the most novel, unfortunately gets the least amount of space and what the authors report is a phenomenological description of the pathway uncovered with no mechanistic studies other than genetic deletion analysis.Thus, the reviewers' general recommendation is that the authors should focus their manuscript and substantially expand either the second story or the third story (in conjunction with the first). If the authors decide to expand the second story, they should describe the effects on novel BGCs and compounds. If the authors decide to expand on the third story, they should further describe the BasR pathway and how it may be activated.1) The authors show that one of the putative Bas1-related TFs, now named BasR, is responsible for the bacterium-induced control but they completely disregard potential function of the second TF, AN8377, based on the apparently unchanged level of H3K9ac in the gene (which is not specifically shown). While AN7174 (BasR) has very restricted distribution in fungi (although there seem to be homologs in Saprolegnia, Malassezia and Ustilago; also according to FungiDB), AN8377 is the more wide-spread factor. The authors do not mention anything about this Myb/SANT domain protein and its function at all. What is shown is a phenomenological description of the pathway uncovered with no mechanistic studies other than genetic deletion analysis. For example, instead of using a rapamycin-deficient mutant of a Streptomyces species that presumably did not cause induction of Gcn5, why wasn't rapamycin simply added to the medium to test whether it was required to elicit the outcome (especially since yeast gcn5 mutants are sensitive to rapamycin). Perhaps this had been done before but then the logic of the experiment with the mutant of an unrelated species is unclear. There is also not direct evidence for BasR regulating either the genes in amino acid metabolism affected by cross-pathway control or the final target genes in clusters, e.g. by binding to motifs in their promoters.

The reviewer criticises two main aspects, to which we answer separately.

First – missing data regarding *AN8377*: Sequence analysis using Blast revealed two putative *bas1* orthologues in *Aspergillus nidulans* – *AN8377* and *basR (AN7174*). Based on our phylogenetic analysis (Figure 6 and Figure 6—figure supplement 1) *AN8377* seems to be more similar to the yeast *bas1* than *basR*. Nevertheless, in contrast to *basR, AN8377* showed no changes in its acetylation pattern during fungal-bacterial co-cultivation (data were added to the subsection “The transcription factor BasR is the central regulatory node of bacteria-triggered SM gene cluster regulation”, first paragraph and Figure 6—figure supplement 1 legend). To further analyse a potential involvement, we generated an *AN8377* deletion strain. Cocultivation experiments with *S. rapamycinicus* revealed no impact of the presence of *AN8377* on the activation of the *ors* gene cluster (new Figure 6—figure supplement 2). Furthermore, we found *basR* to be involved in the activation of the cross-pathway control like the yeast *bas1* and that GcnE is probably needed for its expression. Therefore, we did not further investigate of *AN8377*.

Second – distribution of *AN7174* in fungi: Here, the reviewer is not entirely correct. The only fungal species containing a nearly complete *AN7174* homologue are *Aspergillus versicolor* (score 421/642 (69% )), *Aspergillus sydowii* (score 407/642 (66% )), *Aspergillus calidoustus* (score 355/642 (73% ))*, Aspergillus ochraceoroseus* (score 348/642 (71%))and *Aspergillus rambellii* (score 336/642 (66% )). In all other fungal species (*e.g. Ustilago maydis, Malassezia globosa*) the homologous region is restricted to the DNA-binding Myb domain, which is highly conserved in eukaryotes.

2) If the authors focus their manuscript on the identification of circuits that regulate amino acid availability in Aspergillus, they should give their manuscript a new title. "Chromatin remodeling" specifically refers to the movement of nucleosomes, either sliding, removal or replacement, by a group of large ATPases. The authors are discussing "chromatin modifications" by a lysine acetyltransferase. This is not the same thing as "remodeling". These changes in modifications did not per se result in the identification of BasR, and that BasR directly "regulates" natural product biosynthesis also has not been shown, though it may very well affect this indirectly. Also, BasR is not novel, but distantly related to Bas1/2 in S. cerevisiae.

Thank you for this hint. We changed the title accordingly to **“**Chromatin mapping identifies BasR as the regulatory node of bacteria-induced production of fungal secondary metabolites”. It is correct that, due to its relation to Bas1, BasR is not totally novel, as would not be any Myb-like transcription factor. However, to emphasise the relevance of BasR as a main regulator of natural product biosynthesis in *A. nidulans*, we performed transcriptome analyses with a mutant strain overproducing BasR. Our data reveal that five of the eight secondary metabolite gene clusters differentially acetylated during fungal-bacterial cocultivation are also transcriptionally regulated by BasR. Based on our data the main differences between Bas1 of *S. cerevisiae* and BasR of *A. nidulans* are the different structures of the proteins and the transcriptional regulation of secondary metabolite biosyntheses in *A. nidulans* and thus the regulatory outcome of their activities. Common to both is their involvement in the activation of the cross-pathway control system and the requirement of a functional Gcn5/GcnE.

3) The reviewers appreciate the authors wanting to get good ChIP data on this interaction which really is where the field needs to move. However, the pulldown with antibodies against histone should not pick up any bacterial DNA. Unfortunately, it brought down lots of bacterial DNA. One possibility is that their bacteria have something like Protein A, which would indiscriminately bind the IgG heavy chain. The "fused genome" designation is basically accounting for this contamination which has to have some impact on the results. The reviewers wonder if the authors could have mixed the fungus with cell fragments of the bacterium or some other method to reduce this contamination. The reviewers like the idea of DCA, but are not sure how one can use the data, because of the contamination from the bacterial DNA. We presume that the fungal information is still viable, but would worry that you're introducing biases in library prep, etc., especially when comparing to the fungal samples where 100% of the reads map to Aspergillus. More controls might help. Would this ChIP approach bring down bacterial contamination if other bacteria were used? Maybe some focus on methods would be good to work out a cleaner result.

According to Supplementary file 3, the contamination only makes up to 8-16% of the reads. We are not sure if this qualifies as “lots of”. Anyway, this could be due to various reasons, most likely caused by antibody specificity. The chosen strategy to generate a fused genome accounts for a possible unspecificity by mapping possible contaminated reads to their corresponding genome. Such reads were then deliberately ignored for subsequent processing, most importantly for calculating the normalization factor, which ultimately only takes into account reads mapped to the fungal genome. In conclusion, this strategy reduces bias and enables the comparison of the fungal monoculture with the fungal-bacterial coculture in the best possible way. The success of the strategy becomes evident in Figure 1. The data of cultivations with presence of *A. nidulans* (green and blue lines) are perfectly superimposed to the *Aspergillus* sections but not the *S. rapamycinicus* section. The latter serves as a control. To underline this point further, we have generated MA plots (new Appendix 1—figure 2). They show that both conditions are very similar, with no obvious bias and the loess regressions align with the 0 line. For H3, almost no change is detectable; thus this this data serve as negative control.

[Editors' note: the author responses to the re-review follow.]

The manuscript has been improved but there are some remaining issues that need to be addressed before acceptance, as outlined below:1) Title: change "the regulatory node" to "a key regulatory node" since there are likely other regulatory nodes that contribute to S. rapaminicus-induced SM cluster activation (your data supports this hypothesis: 3/8 differentially acetylated SM gene clusters were not differentially transcribed in response to basR overexpression).

Thank you for the suggestion, the title has been changed to “Chromatin mapping identifies BasR, a key regulator of bacteria-triggered production of fungal secondary metabolites”.

2) The phylogenies of basR depicted on Figure 6 and Figure 6—figure supplement 1 look different. Please explain why.

The phylogenetic tree in Figure 6—figure supplement 1 contains the best non-redundant 50 hits retrieved from the Uniprot Database using the amino acid sequences of BasR, *AN8377* and Bas1 as probes. We thought a more simplified tree makes our point clearer in the printed version of the article. Therefore, in Figure 6, we only showed proteins of the tree that are most similar to BasR, *AN8377* and Bas1. Now, in the caption of Figure 6 we also refer to the more extended tree shown in Figure 6—figure supplement 1.

3) “The changes of H3K14ac in Figure 1 are therefore likely due to nucleosome rearrangements towards the translation start sites (TSS) rather than increased amounts of this modification”: Why does this hypothesis only apply to H3K14ac but not H3K9ac as well?

To explain this aspect, we added the following sentence to the Results: “Such nucleosome rearrangements might represent the prevailing driver of H3K14ac change because at the same time there is a lower overall acetylation level. […] In contrast, changes of H3K9 acetylation levels are stronger, leading to an increase of acetylation despite nucleosome rearrangements.”

4) The authors start the Abstract and Introduction in such a way that suggests that S. rapamycinicus is directly targeting the epigenetic machinery in A. nidulans. Particularly when you say "In line…" in the Abstract, as well as give examples of the secretion of methyltransferases. Is this how you believe the change in chromatin is occurring? There is not enough data presented in this work to support this.

Thank you for this suggestion. At this stage, we are not sure, whether the bacterium directly or indirectly manipulates the fungal chromatin. However, we know that the chromatin manipulation is specific due to this distinct bacterium *Streptomyces rapamyinicus*. We have revised the Abstract and the Introduction according to the suggestions of the reviewer.

5) Abstract, last sentence: I believe you are missing the word "the" between "as…regulatory node".

Changed as recommended.

6) Subsection “Genome-wide profiles of H3K9 and H3K14 acetylation in A. nidulans change upon co-cultivation with S. rapamycinicus”, last paragraph: Could you define what you mean by "chromatin domain" (approx. size)?

After further consideration of this point, we found “acetylation islands” as defined by Roh, Cuddapah and Zhao (2005) to be even more suitable compared to chromatin domain. The relevant section has been altered accordingly.

7) Subsection “The transcription factor BasR is the central regulatory node of bacteria-triggered SM gene cluster regulation”, fourth paragraph: You performed an RNAseq experiment with the overexpression of basR, and describe the changes in secondary metabolite gene cluster expression. Do you see any changes in the other phenotypes observed in the co-culture with S. rapamycinicus? It would be nice to know if there is a decrease in genes associated with nitrogen metabolism and mitochondrial function (Figure 3). This would expand the role of basR. Also, does deletion or overexpression of basR influence gcnE or other members of the Saga/Ada complex expression?

Analysis of the ChIP-seq data revealed differential H3K9 acetylation for genes involved in secondary metabolism, nitrogen metabolism, amino acid metabolism, calcium signaling and asexual development (see Supplementary file 2). We could also show a positive correlation of higher gene expression with increased H3K9 acetylation. Analysis of the RNA-seq data of the *basR* overexpression strain revealed that BasR is not only involved in the transcriptional regulation of several secondary metabolite gene clusters, but also seems to be important for the observed downregulation of genes of the nitrogen metabolism. We have added the corresponding data to the manuscript (Supplementary file 4). The data emphasize the importance of BasR in transducing the bacterial signal(s) in the fungus. We found no influence of BasR on GcnE or other members of the Saga/Ada complex. This is not surprising since GcnE is not differentially expressed during the *A. nidulans – S. rapamycinicus* co-cultivation. We assume that the activity of GcnE is not transcriptionally regulated under the conditions applied in our experiments.

8) Figure 5: Here you illustrate the levels of expression of the ors cluster and basR, as well as the relative levels of the ors cluster products. You mention the "leakiness" of the tet^On^ promoter, and we can see an increase in expression of basR grown without doxycycline. This level of expression looks similar to that in the co-culture, and yet do not see an increase above what is typically seen of the ors cluster. Do you have a hypothesis as to why this is?

It is true, that the difference of the *basR* gene expression between the *A. nidulans* – *S. rapamycinicus* co-cultivation and the non-induced *basR* overexpression is not high. However, in the latter we did not detect orsellinic acid or derivatives thereof (see Figure 5). This finding suggests that under low expression conditions we need an additional, yet unknown signal from the streptomycete. This appears not to be necessary anymore under artificial high BasR overexpression conditions when we see both high expression of *ors* genes and production of orsellinic acid and derivatives. The induction of the *ors* gene cluster in the induced *tet*^On^-*basR* mutant is slower compared to that observed in the co-cultivation of *A. nidulans* with *S. rapamycinicus*. In the wild type, the *ors* gene cluster is already highly expressed after 3 hours of co-cultivation, whereas the overexpression mutant needs six hours after induction by doxycycline to reach a comparable expression level. It is conceivable, that we missed the highest peak of *basR* expression at the analyzed time points. It is also likely that BasR is a member of a protein complex. Thus, overproduction of BasR can lead to a certain overproduction of orsellinic acid, but then other factors of the complex might become limiting.

9) Subsection “The presence of BasR in fungal species allows forecasting the inducibility of ors-like gene clusters by S. rapamycinicus”: Earlier in the manuscript you mention the "leakiness" of the tet^On^ promoter when overexpressing basR in A. nidulans. Is this the case in A. sydowii? If not, it would be really nice to see the loss of ors products in the co-culture of S. rapacycinicus and A. sydowii tet^On^-basR without doxycycline, demonstrating the conserved role of basR in the interaction of the two microbes.

Unfortunately, the used *tet*^On^-system showed similar leakiness in *A. sydowii* and *A. nidulans*. Integrating gene constructs in *A. sydowii* by homologous recombination is very difficult. Therefore, the inducible *tet*^On^construct was integrated ectopically into the genome (see Figure 6—figure supplement 3). Due to this ectopical integration cocultivated *S. rapamycinicus* and *A. sydowii tet*^On^-*basR* would still lead to the production of orsellinic acid and derivatives even in the absence of doxycycline, since expression of the native *basR* in *A. sydowii* remains unaffected. However, we expect that this cultivation condition would bring the same outcome as in the *A. nidulans tet*^On^-*basR* strain that has, of course, the *tet*^On^-*basR* cassette integrated at the *basR* locus and thus replaced the endogenous *basR* gene. This is easily achievable in *A. nidulans*. As expected, cocultivation of the *A. nidulans tet*^On^-*basR* strain with *S. rapamycinicus* in the absence of doxycycline did not lead to the production of any orsellinic acid (see Figure 5).

10) How was the anti-H3 antibody validated? This particular antibody is great for western blots but does not always work for ChIP (see ENCODE histone antibody database, http://compbio.med.harvard.edu/antibodies/targets/12 and the histone validation service by the Strahl lab and company: http://www.histoneantibodies.com/). In our hands this H3 antibody does not work for ChIP. The same concern applies to the anti-H3K9ac and anti-H3K914ac antibodies; Active Motif 39161 does not show up as a validated ChIP antibody in either database, and no catalog number is given for the second antibody.

We have validated the anti-H3 antibody (ab1791) by comparing H3 ChIP efficiency for nucleosomes that were simultaneously mapped for nucleosome positioning. This was carried out in the well-studied bidirectional promoter of the penicillin biosynthesis gene cluster of *A. nidulans*, in which shifts of the environmental pH turn on or off the expression of these genes (*A. nidulans acvA-ipnA* promoter; data not shown). Furthermore, the anti-histone H3 antibody was already used for chromatin immunoprecipitation experiments in *A. nidulans* by the Strauss group in 2008 (Reyes-Dominguez et al., 2008), where it was used to calculate the ratio between acetylated H3 and total H3 of the bidirectional promoter of the *prnD-prnB* genes in *A. nidulans*. Moreover, the reviewer might have overlooked that this antibody is in fact labeled as “validated” in the ENCODE database.

Antibody 39161 (Active Motif) was not used in this work and erroneously mentioned in the Materials and methods section. We are sorry for this mistake and corrected this error. The used anti-H3K9acetyl antibody is a rabbit polyclonal antibody from Abcam (ab4441). Previously, this antibody was successfully used for ChIP analysis by our group (Nützmann et al., 2011). The anti-acetyl-histone H3 (Lys14) is a rabbit polyclonal antibody from Merck (07-353). We added the catalog number for this antibody to the manuscript. In the ENCODE database this antibody was validated as being suitable for ChIP-seq approaches.

11) The model figure is interesting but some of the interactions shown have not been established. The authors show arrows implying that BasR directly regulates the clusters but that has not been shown by experiment. Are binding sites found in the promoters of cluster genes?

Thank you for the comment. The intention of our model figure is to show all known factors involved in the *S. rapamycinicus* – *A. nidulans* interaction, with special emphasis on the BasR-regulated transcription of several secondary metabolite gene clusters, which is clearly supported by our data. We fully agree with the reviewer. At this stage, we cannot exclude that differential expression of the biosynthetic gene clusters is not directly controlled by binding of BasR to the promoters or by another transcription factor that is activated by BasR. To avoid any misunderstanding we have edited the legend to Figure 8:

“Model of the *S. rapamycinicus – A. nidulans* interaction. Co-cultivation leads to activation of the *basR* gene. […] The involvement of AdaB and GcnE of the Saga/Ada complex has been experimentally proven (Nützmann et al., 2011).”

We agree that it would be interesting to identify BasR-binding sites and unravel its regulon, but this would require additional ChIP/ChIP-seq experiments, which we feel go beyond the aims of this study.

12) There is still relatively little mechanistic information about the regulatory cascade from stressor to induction of cluster transcription. How the switch from normal to imbalanced or depleted nitrogen occurs in presence of Streptomyces is the big question, and how GcnE and BasR are linked in Aspergillus is still not clear.

Of course, we agree that we did not show the mechanism(s) how the bacterial signal(s) activating the fungal *ors* gene cluster are transmitted in the fungus. However, we analyzed the genome-wide chromatin acetylation changes during the specific interplay between *A. nidulans* and *S. rapamycinicus*. Based on these data, we found that the fungal chromatin landscape is drastically altered as a result of a functional interaction between the two organisms. We identified a fungal transcriptional regulator required for transducing the bacterial signal(s) which led to the activation of secondary metabolite gene clusters. We found that manipulation of the identified regulator is sufficient to phenocopy the effect of the bacterium on the fungus: overexpressing the corresponding *basR* gene activates orsellinic acid production in absence of the bacterium, while its deletion renders the fungus unable to receive the bacterial instructions and activate the silent SM cluster.

Furthermore, we found that the regulator is functionally conserved in other *Aspergillus* sp., e.g. in *Aspergillus sydowii*, where we also observe the intimate bacterial-fungal interaction. Thus, the presence of the regulator might allow to forecast microbial interaction partners. Overall, our results suggest that key regulatory elements which serve to ‘translate’ bacterial signals into regulation of SM gene clusters, can be manipulated to activate silent secondary metabolite clusters in fungi.

13) Subsection “Increased gene expression directly correlates with histone H3K9 acetylation”: The low correlation between active gene transcription and acetylation at H3K14 confirmed earlier results – a citation is needed here.

Thank you for this remark. We added the references Reyes-Dominguez et al. (2008) and Nützmann et al. (2011).

14) "did not affect the induction of the ors gene cluster, but on the other hand the artificial inducer of the CPC system 3-AT DOES (Sachs, 1996), it…" – this sentence needs a "does" or something similar to make it clear.

Changed as requested.